# Increasing Liquid State Machine Performance with Edge-of-Chaos Dynamics Organized by Astrocyte-modulated Plasticity

**Vladimir A. Ivanov**
Department of Computer Science
Rutgers University
Piscataway, NJ
`vladimir.ivanov@rutgers.edu`

**Konstantinos P. Michmizos**
Department of Computer Science
Rutgers University
Piscataway, NJ
`michmizos@cs.rutgers.edu`

## Abstract

The liquid state machine (LSM) combines low training complexity and biological plausibility, which has made it an attractive machine learning framework for edge and neuromorphic computing paradigms. Originally proposed as a model of brain computation, the LSM tunes its internal weights without backpropagation of gradients, which results in lower performance compared to multi-layer neural networks. Recent findings in neuroscience suggest that astrocytes, a long-neglected non-neuronal brain cell, modulate synaptic plasticity and brain dynamics, tuning brain networks to the vicinity of the computationally optimal critical phase transition between order and chaos. Inspired by this disruptive understanding of how brain networks self-tune, we propose the neuron-astrocyte liquid state machine (NALSM)[1] that addresses under-performance through self-organized near-critical dynamics. Similar to its biological counterpart, the astrocyte model integrates neuronal activity and provides global feedback to spike-timing-dependent plasticity (STDP), which self-organizes NALSM dynamics around a critical branching factor that is associated with the edge-of-chaos. We demonstrate that NALSM achieves state-of-the-art accuracy versus comparable LSM methods, without the need for data-specific hand-tuning. With a top accuracy of $97.61\%$ on MNIST, $97.51\%$ on N-MNIST, and $85.84\%$ on Fashion-MNIST, NALSM achieved comparable performance to current fully-connected multi-layer spiking neural networks trained via backpropagation. Our findings suggest that the further development of brain-inspired machine learning methods has the potential to reach the performance of deep learning, with the added benefits of supporting robust and energy-efficient neuromorphic computing on the edge.

## 1  Introduction

With the recent rise of neuromorphic [1–4] and edge computing [5, 6], the liquid state machine (LSM) learning framework [7] has become an attractive alternative [8–11] to deep neural networks owing to its compatibility with energy-efficient neuromorphic hardware [12–14] and inherently low training complexity. Originally proposed as a biologically plausible model of learning, LSMs avoid training via backpropagation by using a sparse, recurrent, spiking neural network (liquid) with fixed synaptic connection weights to project inputs into a high dimensional space from which a single neural layer can learn the correct outputs. Yet, these advantages over deep networks come at the expense of 1) sub-par accuracy and 2) extensive data-specific hand-tuning of liquid weights. Interestingly, these

---

[1]Code and data available at `https://github.com/combra-lab/NALSM`

35th Conference on Neural Information Processing Systems (NeurIPS 2021).

two limitations have been targeted by several studies that tackle one [15, 16] or the other [17, 18], but not both. This has limited the widespread use of LSMs in real-world applications [8]. In that sense, there is an unmet need for a unified, brain-inspired approach that is directly applicable to the emerging neuromorphic and edge computing technologies, facilitating them to go mainstream.

As a general heuristic, LSM accuracy is maximized when LSM dynamics are positioned at the edge-of-chaos [19–21] and specifically in the vicinity of a critical phase transition [22–25] that separates: 1) the sub-critical phase, where network activity decays, and 2) the super-critical (chaotic) phase, where network activity gets exponentially amplified. Strikingly, brain networks have also been found to operate near a critical phase transition [26–28] that is modeled as a branching process [25, 26]. Current LSM tuning methods organize network dynamics at the critical branching factor by adding forward and backward communication channels on top of the liquid [15, 16]. This, however, results in significant increases in training complexity and violates the LSM's brain-inspired self-organization principles. For example, these methods lack local plasticity rules that are widely observed in the brain and considered a key component for both biological [29] and neuromorphic learning [3, 2, 4]. A particular local learning rule, spike-timing-dependent plasticity (STDP), is known to improve LSM accuracy [17, 18]. Yet, current methods of incorporating STDP into LSMs further exacerbate the limitations of data-specific hand-tuning as they require additional mechanisms to compensate for the STDP-imposed saturation of synaptic weights [17, 30–33]. This signifies the scarcity of LSM tuning methods that are both computationally efficient and data-independent.

A long-neglected non-neuronal cell in the brain, astrocytes, is now known to play key roles in modulating brain networks [34–39], from modifying synaptic plasticity [40–42] to facilitating switching between cognitive states [43–46] that have been linked to a narrow spectrum of dynamics around the critical phase transition [47–51]. The mechanisms that astrocytes use to modulate neurons include the integration of the activity of thousands of synapses into a slow intracellular continuous signal that feeds back to neurons by affecting their synaptic plasticity [52–55, 42]. The unique spatio-temporal attributes [56, 57] identified in astrocytes align well with the brain's remarkable ability to self-organize its massive and highly recursive networks near criticality. That is why astrocytes present a fascinating possibility of forming a unified feedback modulation mechanism required to improve baseline LSM accuracy while eliminating data-specific hand-tuning.

Here, we propose the neuron-astrocyte liquid state machine (NALSM), where a biologically inspired astrocyte model organized liquid dynamics near a critical phase transition, by modulating STDP. We show that NALSM combined the computational benefits of both STDP and critical branching dynamics by demonstrating its accuracy advantage compared to other LSM methods on two datasets: 1) MNIST [58], and 2) N-MNIST [59]. We demonstrate that, similar to its biological counterpart that handles new and unstructured information with robustness and versatility, NALSM maintains the state-of-the-art LSM performance without re-tuning training parameters for each tested dataset. We also show that a NALSM with a large enough liquid can attain comparable accuracy to fully-connected multi-layer spiking neural networks trained via backpropagation on 1) MNIST [58], 2) N-MNIST [59], as well as 3) Fashion-MNIST [60]. Our results suggest that the under-performance and high training difficulty of current neuromorphic methods can be addressed by harvesting neuroscience knowledge and further translating biological principles to computational mechanisms.

## 2 Methods

### 2.1 The neuron-astrocyte liquid state machine

To construct the NALSM, we started with a baseline LSM model consisting of 2 layers: 1) a spiking liquid, and 2) a linear output layer. Next, we added STDP to the LSM liquid, forming the LSM+STDP model. We developed a biologically faithful leaky-integrate-and-modulate (LIM) astrocyte model, which we embedded in the LSM+STDP liquid, to form the NALSM. The process is formalized below.

**LSM Model** We implemented the baseline LSM as a 3-dimensional neural network (liquid) consisting of $1,000$ neurons surrounded by 1-dimensional layers of input and output neurons. Number of input neurons was $784$ and $2,312$ for MNIST and N-MNIST, respectively (See Appendix A.1). We used the leaky-integrate-and-fire (LIF) model [3] for input and liquid neurons, modeled as:

$$\frac{dv_i}{dt} = -\frac{1}{\tau_v}v_i(t) + u_i(t) - \theta_i\sigma_i(t) \tag{1}$$

$$u_i(t) = \sum_{j \neq i} w_{ij}(\alpha_u * \sigma_j)(t) + b_i \tag{2}$$

where $v_i$ is the membrane potential and $u_i$ is the synaptic response current of neuron $i$, $\theta_i$ is the membrane potential threshold, $\sigma_i(t) = \sum_k \delta(t - t_i^k)$ is the spike train of neuron $i$ with $t_i^k$ being the time of the $k$-th spike, $w_{ij}$ is the weight connecting neuron $j$ to $i$, $b_i$ is the bias of neuron $i$, and $\alpha_u(t) = \tau_u^{-1} \exp(-t/\tau_u)H(t)$ is the synaptic filter with H(t) being the unit step function (See Appendix A.2). All LIF neurons had a $2\ ms$ absolute refractory period. Liquid neurons were excitatory and inhibitory with $80\%/20\%$ ratio. Input neurons did not have an excitatory/inhibitory distinction and had random excitatory and inhibitory connections to liquid neurons. From here on, we will refer to connections between input neurons and liquid neurons as IL connections, inter-liquid connections as LL, and liquid to output connections as LO. In line with [7], we created LL connections using probabilities based on Euclidean distance, $D(i,j)$, between any two neurons $i, j$:

$$P(i,j) = C \cdot exp\left(-\left(\frac{D(i,j)}{\lambda}\right)^2\right) \tag{3}$$

with closer neurons having higher connection probability. Parameters $C$ and $\lambda$ set the amplitude and horizontal shift, respectively, of the probability distribution (See Appendix A.3). Density of IL connections was 15%. The output layer was a dense layer consisting of 10 linear neurons.

**LSM+STDP Model** We added unsupervised, local learning to the LSM model by letting STDP change each LL and IL connection [61], modeled as:

$$\frac{dw}{dt} = A_+ T_{pre} \sum_o \delta(t - t_{post}^o) - A_- T_{post} \sum_i \delta(t - t_{pre}^i) \tag{4}$$

where $A_+ = A_- = 0.15$ are the potentiation/depression learning rates and $T_{pre}/T_{post}$ are the pre/post-synaptic trace variables, modeled as,

$$\tau_+^* \frac{dT_{pre}}{dt} = -T_{pre} + a_+ \sum_i \delta(t - t_{pre}^i) \tag{5}$$

$$\tau_-^* \frac{dT_{post}}{dt} = -T_{post} + a_- \sum_o \delta(t - t_{post}^o) \tag{6}$$

where $a_+ = a_- = 0.1$ are the discrete contributions of each spike to the trace variable, $\tau_+^* = \tau_-^* = 10$ $ms$ are the decay time constants, $t_{pre}^i$ and $t_{post}^o$ are the times of the pre-synaptic and post-synaptic spikes, respectively. We constrained connection weights to: 1) IL: $[-3, 3]$ , 2) excitatory LL: $[0, 3]$, and 3) inhibitory LL: $[-3, 0]$. We used the same STDP parameters for all models and experiments.

**LIM Astrocyte Model** We developed the astrocyte model as a leaky integrator with a continuous output value $A_-^{astro}$, expressed as:

$$\tau_{asto} \frac{dA_-^{astro}}{dt} = -A_-^{astro} + w_{astro} \sum_{i \in N_{liq}} \delta(t - t_i) - w_{astro} \sum_{j \in N_{inp}} \delta(t - t_j) + b_{astro} \tag{7}$$

where $A_-^{astro}$ directly mapped to $A_-$ in equation (4), $b_{astro} = A_+$ adjusted the astrocyte output to the fixed STDP potentiation learning rate, $N_{liq}$ and $N_{inp}$ are the sets of liquid and input neurons, respectively, and $w_{astro}$ set astrocyte responsiveness to network activity (See Appendix A.4). Ignoring the decay and bias terms, the astrocyte model computed the difference in the number of spikes produced by liquid neurons and input neurons. Functionally, this is equivalent to computing the ratio of spikes emitted by the liquid over the input neurons:

$$BF_{proxy}(t) = \frac{\sum_{i \in N_{liq}} \delta(t - t_i)}{\sum_{j \in N_{inp}} \delta(t - t_j)} \tag{8}$$

Specifically, when the liquid produced more spikes than the input neurons, their difference was positive which translated to $BF_{proxy} > 1.0$, and vice versa. This approach to measure liquid dynamics acted as a network level approximation of the branching factor, $\sigma_{BF}$, which is normally

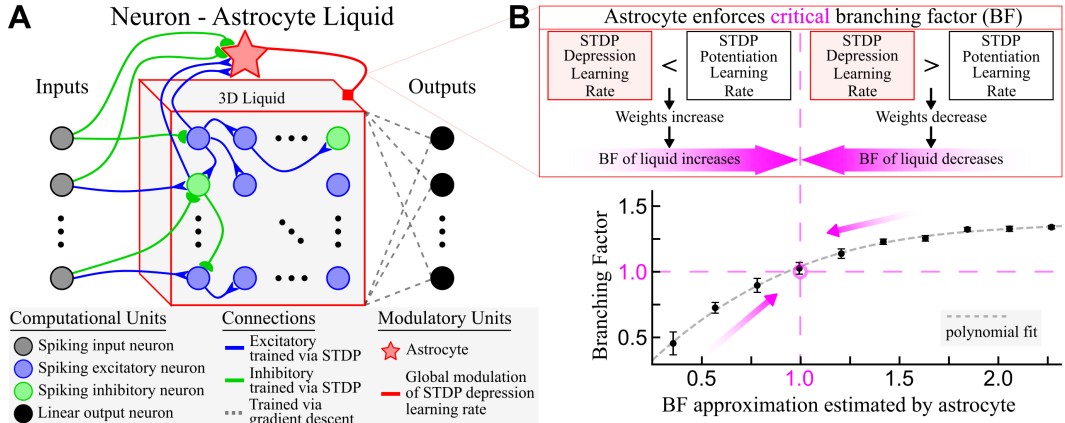

Figure 1: **NALSM architecture and astrocyte modulation of liquid dynamics.** ( **A** ) The neuron-astrocyte liquid was modeled as a 3-dimensional network of excitatory and inhibitory spiking neurons connected with sparse, recurrent connections with spike-timing-dependent plasticity (STDP). Input neurons projected excitatory and inhibitory connections to the liquid. Receiving each liquid neuron's spike count per input sample, a dense linear output layer was trained via gradient descent to classify inputs. ( **B** ) To organize liquid dynamics at the critical branching factor, an astrocyte integrated input and liquid neuron activity and, in turn, set the global STDP depression learning rate. Data points are binned averages over 'BF approximation' metric. Error bars are standard deviation. See Appendix A.10 for polynomial fits.

evaluated for each neuron (See 2.3.1). We empirically confirmed that $BF_{proxy} = 1.0$ aligned with the critical branching factor, $\sigma_{BF} = 1.0$ (Fig. 1 B). Hence, as dynamics became progressively super-critical ($\sigma_{BF} > 1.0$), $BF_{proxy}$ became greater than 1, which caused the LIM astrocyte to increase STDP depression learning rate above the fixed STDP potentiation learning rate ($A^{astro}_{-} \to A_{-} > A_{+}$). This caused STDP to decrease the average weight of LL and IL connections, which decreased number of spikes produced by the liquid and made dynamics less super-critical (Fig. 1 B). The reverse occurred as dynamics became progressively sub-critical. As a result of astrocyte modulation, liquid dynamics oscillated between sub-critical and super-critical until eventual stabilization near the critical branching factor (See Appendix A.4).

**NALSM Model** We completed NALSM by adding the LIM astrocyte to the LSM+STDP model's liquid (Fig. 1 A). As described above, the LIM astrocyte integrated activity from input and liquid neurons, and continuously controlled the STDP depression learning rate.

## 2.2 Training

Model training was done in 3 steps: 1) initialization of IL and LL liquid connections, 2) passing all data through the liquid resulting in liquid neuron spike counts, and 3) training the output layer on the spike counts. The steps are further detailed below.

### 2.2.1 Liquid initialization

**LSM** We initialized all IL and LL connections with a single weight value, maintaining originally defined connection signs [62]. Weights were constant during spike count collection.

**LSM+STDP** We initialized IL and LL connections with STDP by consecutively presenting all the training images to the liquid. Starting with initially maximal connections, $3(-3)$ for excitatory(inhibitory) connections, we let STDP continuously adjusted weights while presenting the liquid with a randomly ordered series of MNIST training image snapshots, each lasting $20\ ms$. For N-MNIST, we randomly sampled each $20\ ms$ snapshot from the $0 - 250\ ms$ range, as a way to account for the variability in the temporal dimension (See 2.3). In each case, we used a total of $50,000$ snapshots, each corresponding to a unique training image. We used STDP only for weight initialization. Initialized weights were fixed during spike count collection.

**NALSM**  We used the LSM+STDP weight initialization process with the exception of added STDP modulation by the LIM astrocyte. We used this set of initialized weights as the starting point for each sample in the spike counting phase, during which astrocyte-modulated STDP continued to adjust synaptic weights to compensate for slight deviations in dynamics caused by each input sample's different level of activity. For each sample, parameters $A_+$ from (4) and $b_{astro}$ from (7) were both initialized to 0.15 and decayed at a rate of 0.99 for the duration of sample input.

### 2.2.2  Output layer training

We assembled spike counts by presenting each sample image to the liquid for 250 $ms$ and counting the number of spikes emitted by each liquid neuron for the full duration of input. We used Adam optimizer to batch train the output layer on spike count vectors by minimizing the cross entropy loss with L2-regularization,

$$\mathcal{L}(y_i, \hat{y}) = -\frac{1}{m}\sum_{i=1}^{m} y_i log(\hat{y}_i) + (1-y_i)log(1-\hat{y}_i) + \frac{\lambda_{reg}}{2m}||W_{out}||_F^2 \qquad (9)$$

where $m = 250$ is the batch size, $W_{out}$ is the output layer weight matrix, $\lambda_{reg} = 5 \times 10^{-10}$ is the regularization hyperparameter, $y_i$ and $\hat{y}_i$ are the normalized vectors denoting the predicted label and the target label, respectively. Prior to training, we initialized output layer weights/biases to 0.0, and the learning rate to 0.1. We trained the output layer until validation accuracy peaked (up to a maximum of 5,000 epochs), at which point we evaluated model test accuracy.

### 2.3  Experiments

We performed all LSM comparison experiments on MNIST and N-MNIST datasets (See A.1). Using 10 randomly generated networks for each dataset, we trained 1) the baseline LSM model, 2) the LSM+STDP model, 3) the NALSM model (See 2.1), and 4) the LSM+AP-STDP model, a method for incorporating STDP in the LSM liquid [17] (See Appendix A.5). First, we evaluated LSM model accuracy with respect to liquid weight, which ranged in $0.4 - 1.2$ for MNIST and $0.8 - 1.35$ for N-MNIST. We used a random seed for each training session. Next, we evaluated corresponding network dynamics of each network/weight combination by measuring the liquid's branching factor on 20 randomly sampled inputs (See 2.3.1). To have comparable results for each network, we trained the remaining models using the same seed that resulted in peak LSM accuracy. For NALSM, we used the same initialization and parameters for all networks and datasets. For LSM+AP-STDP, we hand-tuned STDP control parameters for each network and dataset combination to maximize validation accuracy (See Appendix A.5). Additionally, we trained NALSM on Fashion-MNIST dataset (See A.1) using the same 10 randomly generated networks that we had used for MNIST.

**Sparse neuron-astrocyte connectivity**  We tested NALSM's accuracy as a function of neuron-astrocyte connection density on 3 best performing networks (per dataset). Keeping the proportion of neurons sampled by the astrocyte the same for both input neurons and liquid neurons, we trained NALSM with $10\%, 20\%, 40\%, 60\%$, and $80\%$ neuron-astrocyte density over 3 seeds for each of 3 networks. Regardless of connection sparseness, all IL/LL connections were modulated by $A_-^{astro}$ (See 2.1)

**NALSM with larger liquid sizes**  We tested NALSM performance for larger liquids. For each size, we trained 3 randomly generated networks, each on a random seed. All parameters and initialization were same as for 1,000 neuron liquid. For maximum accuracy, we trained NALSM with an 8,000 neuron liquid. For each dataset, we used 5 randomly generated networks trained on a random seed. Parameter $w_{astro} = 0.0075$ for all datasets. All other parameters and initialization were as before.

### 2.3.1  Branching factor of liquid

To evaluate a liquid's dynamics, we used the network branching factor, $\sigma_{BF}$, which quantifies network information flow amplification/decay. Liquid dynamics are sub-critical, near-critical and super-critical when $\sigma_{BF} < 1.0$, $\sigma_{BF} \approx 1.0$, and $\sigma_{BF} > 1.0$. We calculated $\sigma_{BF}$ as done in [30], with offset $\phi = 0$ and time window $\Delta = 4 \; ms$ as per [26].

### 2.3.2 Kernel quality of liquid

We evaluated liquid 1) linear separation, and 2) generalization capability using methods from [63]. For the MNIST and N-MNIST test sets, we computed the rank of matrix $M$ assembled from $k$ randomly selected spike count vectors, resulting in shape $N_{liq} \times k$. We repeated this on $1,000$ shuffles of spike vectors. For linear separation, we used spike counts from model testing phase. For generalization capability, we added noise to input data and evaluated new spike count vectors (See 2.2.2). For MNIST, we added $\mathcal{N}(0, 125)$ noise to each pixel value. For N-MNIST, we time-shifted each event by $\mathcal{N}(0, 10)$. Taken together across all models and both datasets, we rescaled ranks of each measure to $0 - 1$ range and subtracted measure 1 from measure 2 as in [63]. Due to negative differences, we again rescaled all differences to $0 - 1$.

## 3 Results

### 3.1 Baseline LSM performance

We established a benchmark accuracy for the baseline LSM on MNIST and N-MNIST datasets (See 2.3). We acquired our baseline by averaging over 10 randomly generated liquids with $1,000$ neurons (See 2.1). The LSM achieved a top accuracy of $95.44\%$ ($95.30 \pm 0.11\%$) on MNIST, and $95.35\%$ ($95.02 \pm 0.15\%$) on N-MNIST (See 2.2). For MNIST, this was comparable to the previously reported state-of-the-art LSM accuracy [64], using the same sized liquid. Further, LSM accuracy was very sensitive to the liquid's weight (Fig. 2 A).

### 3.2 LSM performance peaked at the critical branching factor

The peak LSM accuracy on each dataset corresponded to a different liquid synaptic weight. Specifically, there were cases where a liquid with weights tuned for maximum accuracy on MNIST, would catastrophically fail on N-MNIST (Fig. 2 A). Also, LSM accuracy on MNIST plateaued for a wider

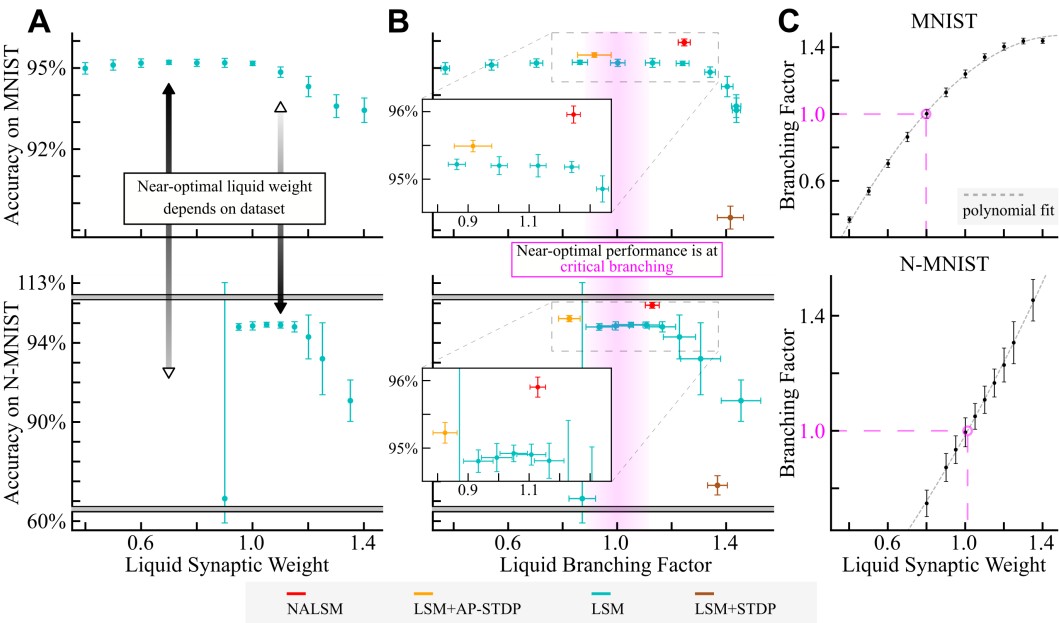

Figure 2: **LSM accuracy depended on liquid weight and dynamics.** ( **A** ) LSM accuracy shown as a function of its liquid weight, averaged over a set of 10 randomly generated networks for MNIST and N-MNIST datasets. ( **B** ) LSM accuracy shown with respect to liquid dynamics set by liquid synaptic weight. For each weight, liquid dynamics were measured and averaged over all 10 networks. Similarly, accuracy and resulting liquid dynamics are shown for each model: 1) NALSM, 2) LSM+AP-STDP, and 3) LSM+STDP. ( **C** ) Liquid dynamics shown with respect to liquid weight, averaged over all 10 networks. Error bars are standard deviation. See Appendix A.10 for polynomial fits.

range of weights than on N-MNIST, which can be attributed to N-MNIST's greater difficulty caused by its variability over the temporal dimension (See Appendix A.1). Taken together, this indicated that LSM training requires extensive hand-tuning of weights for each specific dataset.

Since critical dynamics are well known to result in near-maximum LSM performance [65, 19–21], dataset-specific hand-tuning can be significantly reduced by replacing accuracy with liquid dynamics as the target output of weight tuning. Indeed, LSM accuracy was near-maximum for both datasets, when the liquid's branching factor was in $1.0 - 1.2$ range, or slightly super-critical (Fig. 2 B) (See 2.3.1). This agrees with studies showing that information transfer in finite sized systems peaks at slightly super-critical dynamics [66]. Although each dataset still had different weight ranges corresponding to the critical branching factor, the relationship between liquid dynamics and weight was positive for both datasets (Fig. 2 C). Known to generalize beyond specific datasets [20], this relationship suggested that near-critical dynamics can be organized using STDP, by providing it directional feedback from current liquid dynamics.

### 3.3 Astrocyte-modulated plasticity organized liquid dynamics near criticality

The LIM astrocyte model stabilized liquid dynamics near the critical branching factor as we presented a continuous stream of samples to the neuron-astrocyte liquid (Fig. 2 B). The NALSM's slightly super-critical stabilization suggested that liquid dynamics were at the edge-of-chaos. While chaotic activity is known to correspond to super-critical branching dynamics in some cases [25], such correspondence is not guaranteed. Hence, we examined additional network properties that are necessary and indicative of chaotic activity (See Appendix A.6). Specifically, the astrocyte-modulated liquid had coexistence of small and large synaptic weights (Fig. S2), as well as a balance of excitation and inhibition, both of which are necessary for the existence of chaotic network activity [67, 68]. Further supporting a chaotic activity, the neuron-astrocyte liquid spike activity appeared irregular (Fig. S3). We also performed autocorrelation analysis on liquid neuron spike trains, which further suggested the existence of chaotic activity with a correspondence between edge-of-chaos dynamics and critical branching dynamics (Fig. S4) [69, 70] (See Appendix A.6). Given that liquid dynamics were directly approximated by the LIM astrocyte, which directly controlled the STDP depression learning rate (See 2.1), NALSM required no dataset-specific hand-tuning. As a result, we used the same weight initialization and parameters to benchmark NALSM (See 2.3).

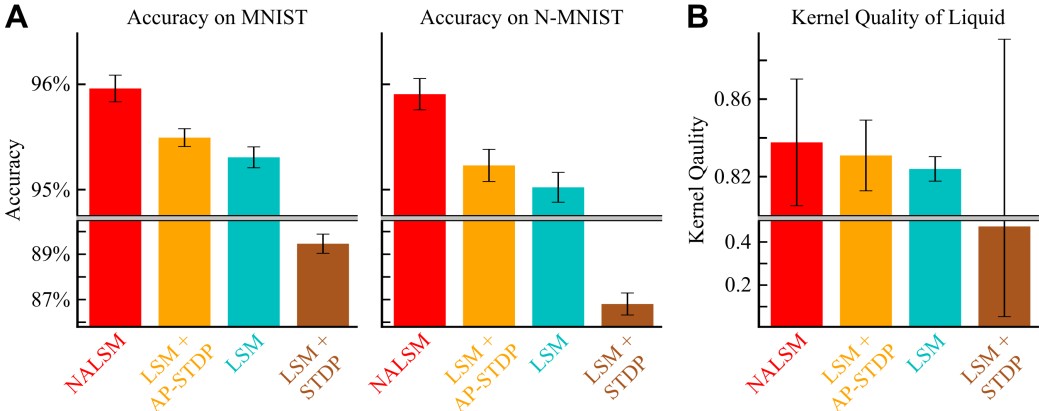

Figure 3: **Comparison of model accuracy and liquid computational capacity.** ( **A** ) Accuracy performance of the proposed NALSM model was compared, on MNIST and N-MNIST, against 3 related models: 1) the baseline LSM, 2) LSM with activity-based STDP (LSM+AP-STDP), and 3) LSM with unregulated STDP (LSM+STDP). For each dataset and model, accuracy was evaluated using 10 randomly generated networks, each of which was trained on a random seed. This set of 10 seeds was used for all models. ( **B** ) Computational capacity of each model was measured using a kernel quality metric that encompassed the linear separation and generalization capability of the liquid (See 2.3.2). Error bars are standard deviation.

### 3.4 Benchmarking NALSM performance on MNIST and N-MNIST

On both datasets, NALSM achieved superior performance to comparable LSM models of the same size. Using $1,000$ liquid neurons, NALSM achieved a top accuracy of $96.15\%$ ($95.96 \pm 0.13\%$) on MNIST and $96.13\%$ ($95.90 \pm 0.16\%$) on N-MNIST; outperforming LSM model's top accuracy by $0.71\%$ on MNIST and $0.78\%$ on N-MNIST (Fig. 3 A). We also compared NALSM to a state-of-the-art LSM STDP method, AP-STDP (See 2.3). The LSM+AP-STDP model required more extensive dataset-specific hand-tuning than the baseline LSM due to its additional STDP control parameters (See Appendix A.5). Resulting in top accuracy of $95.62\%$ ($95.49 \pm 0.09\%$) and $95.43\%$ ($95.23 \pm 0.16\%$), the LSM+AP-STDP model was superseded by NALSM by $0.53\%$ and $0.70\%$ on MNIST and N-MNIST, respectively. As a control measure, we also trained a LSM with unregulated STDP (See 2.1). The LSM+STDP model significantly under-performed compared to all other models achieving a top accuracy of $90.52\%$ ($89.47 \pm 0.45\%$) on MNIST and $87.71\%$ ($86.80 \pm 0.51\%$) on N-MNIST. We attributed this under-performance to the LSM+STDP liquid's excessive super-critical dynamics (Fig. 2 B) which are well known to decrease liquid computational capacity [65, 66].

The NALSM had the most robust accuracy performance across the two datasets out of all the compared LSM models. With no dataset-specific tuning, NALSM's average accuracy on N-MNIST was lower than the accuracy on MNIST by only $-0.05\%$. This was $5 - 50$ times less than for the LSM+AP-STDP ($-0.26\%$), LSM ($-0.29\%$), and LSM+STDP ($-2.66\%$) models.

We attributed the NALSM's performance advantage to the improved computational properties of its liquid. For both tested datasets, the NALSM achieved slightly super-critical branching dynamics where baseline LSM performance peaked right before it started to decline with increasing super-critical dynamics (Fig. 2 B). This suggested that NALSM's performance advantage, compared to a LSM with similar dynamics, was due to the addition of astrocyte-modulated STDP (See 2.1, 2.2.1). While LSM+AP-STDP and LSM+STDP models also had STDP, their lower performance can be explained by their excessively sub-critical and super-critical dynamics, respectively (Fig. 2 B). We further confirmed that NALSM's increased performance resulted from the improved computational properties of its liquid by measuring each model's liquid kernel quality. This encompassed both the linear separation and generalization capability of the liquid (See 2.3.2). Higher model accuracy corresponded to higher kernel quality for all $4$ models (Fig. 3 B). This is a further indication that near-critical dynamics and astrocyte-modulated STDP contributed to the NALSM's performance increase.

### 3.5 NALSM maintained performance with sparse neuron-astrocyte connectivity

The NALSM maintained its accuracy advantage even with neuron-astrocyte connection densities as low as $10\%$ (Fig. 4). In the brain, astrocytes contact only approximately $65\%$ of all synapses in their surroundings [71]. We tested NALSM performance as a function of neuron-astrocyte connection density (See 2.3). The NALSM mean accuracy decreased marginally with increasingly sparse connectivity, while variability in performance was minimal across densities. At $10\%$ connectivity, average NALSM accuracy decreased by $0.36\%$ for MNIST and $0.14\%$ for N-MNIST compared to $100\%$ connection density. In both cases, average NALSM accuracy was still above LSM and LSM+APSTDP average accuracy.

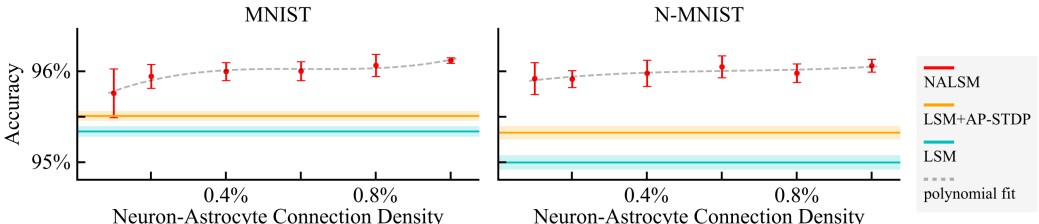

Figure 4: **NALSM maintains accuracy advantage with sparse neuron-astrocyte connectivity.** For MNIST and N-MNIST, NALSM accuracy was evaluated with respect to neuron-astrocyte connection density. For each density, NALSM performance was compared to LSM and LSM+AP-STDP average accuracy. NALSM data points are average values over $9$ experiments (3 networks $\times$ 3 seeds). Error bars and shaded areas are standard deviation. See Appendix A.10 for polynomial fits.

Table 1: Comparison to brain-inspired and fully-connected multi-layer spiking neural networks.

| Model | Layers | Learning Method | Accuracy |
|---|---|---|---|
| **Dataset: MNIST** | | | |
| Unsupervised-SNN [75] | 2 | STDP | 95% |
| Multi-liquid LSM [64] | 2 | GD on last layer | 95.5% |
| **NALSM1000** | **2** | **astro-STDP, GD on last layer** | **96.15%** |
| LIF-BA [73] | 3 | Broadcast feedback alignment | 97.09% |
| Temporal SNN [76] | 2 | Temporal backpropagation | 97.2% |
| STiDi-BP [77] | 2 | Backpropagation | 97.4% |
| **NALSM8000** | **2** | **astro-STDP, GD on last layer** | **97.61%** |
| SN [78] | 3 | Backpropagation | 97.93% |
| GLSNN [72] | 4 | Global feedback alignment, STDP | 98.62% |
| Balance-SNN [74] | 2 | Equi-prop, STDP, STP | 98.64% |
| BPSNN [79] | 3 | Backpropagation | 98.88% |
| STBP [80] | 2 | Spatial and temporal backpropagation | 98.89% |
| **Dataset: N-MNIST** | | | |
| DECOLLE [81] | 3 | Backpropagation | 96% |
| **NALSM1000** | **2** | **astro-STDP, GD on last layer** | **96.13%** |
| AER-SNN [82] | 2 | Backpropagation | 96.3% |
| **NALSM8000** | **2** | **astro-STDP, GD on last layer** | **97.51%** |
| BPSNN [79] | 3 | Backpropagation | 98.74% |
| STBP [80] | 2 | Spatial and temporal backpropagation | 98.78% |
| SLAYER [83] | 3 | Backpropagation | 98.89% |
| **Dataset: Fashion-MNIST** | | | |
| VPSNN [84] | 2 | Equi-prop, STDP | 82.69% |
| **NALSM1000** | **2** | **astro-STDP, GD on last layer** | **83.54%** |
| Unsupervised-SNN [85] | 2 | STDP | 85.31% |
| **NALSM8000** | **2** | **astro-STDP, GD on last layer** | **85.84%** |
| BS4NN [86] | 2 | Temporal backpropagation | 87.3% |
| GLSNN [72] | 4 | Global feedback alignment, STDP | 89.05% |

*GD: gradient descent

### 3.6 Larger liquids increased NALSM accuracy

The NALSM accuracy improved with increased liquid size, saturating at approximately $8,000$ neurons (See Appendix A.7). NALSM8000 achieved a top accuracy of $97.61\%$ ($97.49 \pm 0.11\%$) on MNIST, $97.51\%$ ($97.42 \pm 0.07\%$) on N-MNIST, and $85.84\%$ ($85.61 \pm 0.18\%$) on Fashion-MNIST. Compared to previously reported benchmarks on MNIST and Fashion-MNIST, the NALSM8000 outperformed all brain-inspired learning methods that do not use backpropagation of gradients or its approximation through feedback alignment [72, 73], with the exception of [74] for MNIST. While [74] demonstrated that a fully-connected 2-layer spiking network can achieve high accuracy through a combination of biologically-plausible plasticity rules, it is not clear how such an approach would scale to more layers without some form of backpropagation. Conversely, multi-layered LSMs have been shown to work without backpropagation [8, 10]. Further, NALSM8000 used approximately $1/3$ ($\approx 1,199,407 \pm 453$) of number of trainable(plastic) connections as in [74]. Compared to top accuracies reported for fully-connected multi-layered spiking neural networks trained with backpropagation, the NALSM8000 achieved comparable performance on all datasets; outperforming multiple reported results on MNIST and N-MNIST (Table 1) (See Appendix A.8).

## 4 Discussion and Broader Impact

Ironically, LSMs are one of the most brain-like and at the same time one of the most difficult to train learning models. Here, we proposed an astrocyte model that merged critical branching dynamics and STDP into a single liquid, thereby simultaneously improving LSM performance and decreasing data-specific tuning. We showed that the synergy of STDP and near-critical branching dynamics

improved the computational capacity of the liquid, which translated to better than state-of-the-art LSM accuracy on MNIST and N-MNIST, and do so with minimal added computational cost (See Appendix A.9). Our results indicate that, given a large enough liquid, NALSM performance compares to current fully-connected multi-layer spiking neural networks trained via backpropagation.

The reported narrowing of the performance gap between brain-inspired LSM and deep networks suggests that studying the interaction among the brain's computational principles can help our learning models to reach human-like performance. Indeed, our results demonstrate that the synergy of brain-inspired astrocyte-modulated STDP and near-critical dynamics resulted in the superior performance of NALSM compared to 1) a LSM with critical dynamics but without STDP, and 2) a LSM with STDP, but without critical dynamics. Aligning with other studies showing that liquid topology impacts LSM accuracy [87], we also showed that a brain-inspired, sparse, 3D-distance-based network architecture can improve the computational capacity of a single liquid. Specifically, our baseline 3D LSM achieved comparable accuracy to the multi-liquid LSM [64], which improved performance of a single dimensionless liquid by partitioning it into multiple liquids. While we demonstrated NALSM performance using only the 3D-distance-based network architecture, our proposed astrocyte modulation method does not depend on network topology and, therefore, is applicable to other types of topology. In fact, our approach is also extendable to multi-liquid architectures and other local plasticity rules that follow STDP's separation of potentiation and depression components.

The astrocyte-modulated LSM learning framework is also compatible with the emerging neuromorphic hardware. This is because the gradient descent that we used for training the linear output can be replaced by a single-layer spike based learning rule [88–90]. This makes NALSM compatible with neuromorphic hardware, exploiting in full its advantages [91]. For example, our method can leverage even further the energy efficiency of neuromorphic chips, by virtue of its low spiking rates. In line with biological ranges [92], NALSM had spiking rates that ranged from $12\,Hz$ to $37\,Hz$, depending on the input sample. These rates can be reduced further, by modifying input encoding, since liquid spiking rates are directly adjusted by the astrocyte based on input spiking rates (Fig. 1).

Here, we demonstrated a possible connection between the near-critical branching dynamics of the NALSM liquid and the edge-of-chaos transition (See Appendix A.6). The critical branching transition has been extensively used to model critical dynamics in brain networks [27, 26]. Focusing on the computational benefits of criticality, machine learning has mostly examined network dynamics at the edge-of-chaos transition. Although the presence of one transition does not guarantee the existence of the other, both transitions are well connected to the same result, an improved computational performance [25]. Indeed, the computational performance of systems poised at a critical phase transition has been widely studied both experimentally [22] and theoretically [23], and are well-connected to both edge-of-chaos [19, 20] and critical branching transitions [25, 15]. Networks operating at near-criticality are believed to have simultaneous access to the computational properties (learning and memory) of both phases, which results in 1) maximizing their information processing capacity [22], 2) optimizing their dynamical range [93, 24], and 3) expanding their number of metastable states [25]. Hence, it is not surprising that the NALSM's astrocyte imposed near-critical branching dynamics resulted in improved accuracy and generalization capabilities as observed in LSMs with edge-of-chaos dynamics [19, 20], while adding the benefit of a neuromorphic compatibility and self-organized criticality.

Our work shows how insights from modern cellular neuroscience can synergize with neuromorphic computing, and lead to novel intelligent systems, spurring the dialogue between artificial intelligence and brain sciences. Indeed, given that the known neuronal mechanisms are too slow and uncoordinated in the brain to modulate STDP [31, 94, 32], it is an open question how neurons modulate synaptic plasticity. Our demonstration that the distinct temporal and spatial mechanisms of astrocytes may modulate STDP and subsequently regulate network dynamics, questions the neuron as the only processing unit in the brain [95–97]. In that sense, it helps in dismantling the 100-year old dogma that "brain = neurons", and tackle the absence of astrocytes in both prevailing computational hypotheses on how the brain learns and efforts to translate such knowledge to effective models of intelligence.

By showing how astrocyte-modulated STDP can maximize computational performance near criticality, we aimed to broaden the applicability of the LSM to complex spatio-temporal problems that require integration of data over multiple sources and time-scales, thereby, making LSMs suitable for real-life applications of edge computing. Our so far results suggest that this is a direction worth pursuing.

## Acknowledgements

This work is supported by the National Center for Medical Rehabilitation Research (NIH/NICHD) $K12HD093427$ Grant and by the Rutgers Office of Research and Innovation. Any findings, conclusions, and opinions expressed in this material are those of the authors and do not necessarily reflect the views of the NIH or Rutgers University.

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
