# A Appendix

## A.1 Datasets

We tested models on MNIST [1], its temporal, event-driven version, N-MNIST [2], and Fashion-MNIST [3]. We modified the original $60,000/10,000$ train/test split to $50,000/10,000/10,000$ train/validate/test split, by partitioning away the last $10,000$ training samples to the validation set. We normalized and rescaled each $28 \times 28$ MNIST and Fashion-MNIST image to $0 - 1$ range, which we Poisson encoded into the spiking activity of input neurons. For N-MNIST, we treated all discrete events the same way and transformed each image into $300 \times 68 \times 34$ matrix, with the first dimension being temporal. Using first 250 timesteps, we converted each event at each timestep into a spike in the corresponding input neuron.

## A.2 Neuron Parameters

The LIF neuron parameters we used in all networks are shown in Table S1.

## A.3 Liquid Connectivity Parameters

The parameters we used in the distance based connection probability function, (3), depended on the connection type. Connection types were determined by the pre- and post-synaptic neurons, which resulted in 4 types of connections:

1. $EE$: excitatory to excitatory
2. $EI$: excitatory to inhibitory
3. $II$: inhibitory to inhibitory
4. $IE$: inhibitory to excitatory

For each connection type $[EE, EI, II, IE]$, parameter $C$ values were $[0.2, 0.1, 0.3, 0.05]$. For all connection types, $\lambda = 3.0$.

## A.4 Neuron-astrocyte connection weight

The weight of neuron-astrocyte connections, $w_{astro}$ in (7), impacted both liquid dynamics and NALSM accuracy. Controlling the responsiveness of the LIM astrocyte to liquid neuron activity, larger $w_{astro}$ resulted in lower branching factor, and vice versa. For both datasets, accuracy peaked in the vicinity of $w_{astro} = 0.01$ with slightly super-critical branching factor of $\approx 1.3$ for MNIST and $\approx 1.2$ for N-MNIST (Fig. S1).

## A.5 LSM+AP-STDP model

We implemented AP-STDP from [4] on top of LSM+STDP model by making STDP weight changes conditionally dependent on neuronal activity. Specifically, we implemented rule (4) from [4], with $p = 1.0$. The spiking rate of each neuron $i$, $C_i$, was approximated using (3) from [4], with $\tau_C = 1000$ $ms$. Parameters $C_\theta$ and $\Delta C$ set the neuronal activity range in which STDP changes were enforced. We hand-tuned parameters $C_\theta$ and $\Delta C$ for each specific network and dataset to maximize the validation accuracy of LSM+AP-STDP model. We used the same initialization process as was used for LSM+STDP model, with two exceptions 1) weights were set to 1.0 prior to initialization, and 2)

Table S1: LIF neuron parameters

| Parameter name | Description | Value |
|---|---|---|
| $\theta$ | membrane potential threshold | 20.0 |
| $\tau_v$ | membrane potential time constant | 64.0 |
| $\tau_u$ | synaptic conductance time constant | 1.0 |
| $b$ | membrane potential bias | 0.0 |

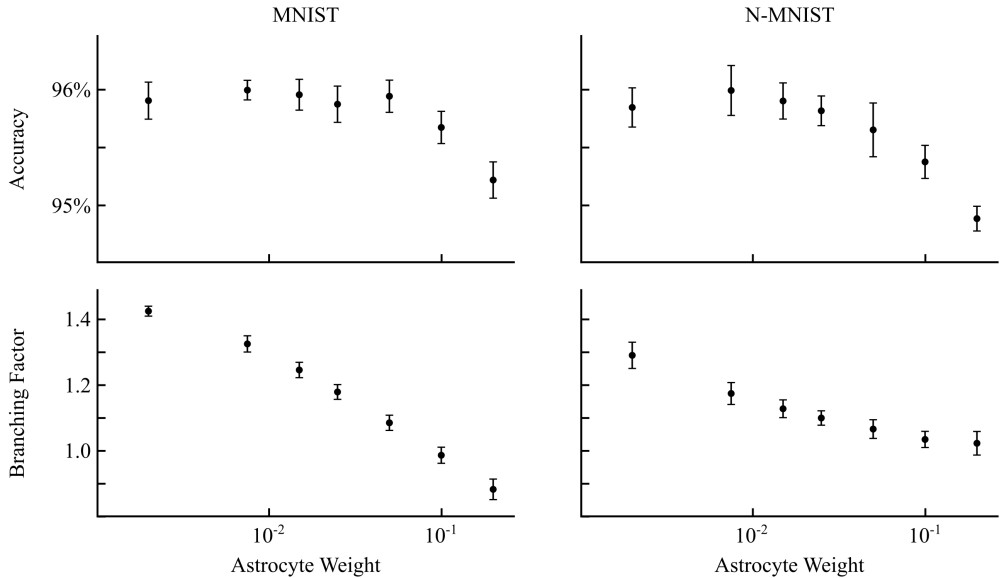

Figure S1: **Neuron-astrocyte connection weight impacts liquid dynamics and NALSM accuracy.** ( **Top** ) NALSM accuracy shown with respect to neuron-astrocyte connection weight for MNIST and N-MNIST. ( **Bottom** ) NALSM liquid dynamics shown as a function of neuron-astrocyte connection weight for MNIST and N-MNIST. Data points are averaged over 10 random networks. Error bars are standard deviation.

STDP synaptic weight changes were conditioned on neuron activity ranges using $C_\theta$ and $\Delta C$. As with LSM+STDP model, the liquid's weights were fixed during spike generation phase.

### A.6 Evidence for edge-of-chaos dynamics in NALSM

Here, we provide evidence suggesting that NALSM's slightly super-critical branching dynamics (Fig. 2) corresponded to the edge-of-chaos. First, NALSM exhibited coexistence of small and large synaptic weights, which is necessary for chaotic activity in spiking networks [5]. NALSM had concentrations of near-maximum excitatory weights and near-zero weights, with weights also covering the full range in between these extremes. Inhibitory weights exhibit the same kind of bimodal distribution (Fig. S2).

Second, NALSM exhibited excitation/inhibition (E/I) balance, which is thought to be necessary for existence of deterministic chaos [6, 7]. We used three different methods to evaluate E/I balance. First, we confirmed synaptic weight E/I balance, $W_{E/I}$, in initialized NALSM liquids, which was found to align with edge-of-chaos dynamics in [7] and was evaluated as:

$$W_{E/I} = \frac{n_{w>0} - n_{w<0}}{n_{w \neq 0}} \tag{10}$$

where $n_{w>0}$, $n_{w<0}$, and $n_{w \neq 0}$ are total number of IL and LL synaptic weights that are positive, negative, and non-zero, respectively. Indicative of E/I balance, we obtained $W_{E/I} = -0.0029 \pm 0.018$ averaged over all NALSM initializations on both MNIST and N-MNIST ($W_{E/I}$ ranges from $-1$ to $1$, with $0$ representing perfect E/I balance). Second, we measured the difference in spiking rates between liquid excitatory and liquid inhibitory neuron populations by evaluating:

$$f_{E/I} = \frac{|\hat{f}_e - \hat{f}_i|}{\hat{f}_l} \tag{11}$$

where $\hat{f}_e$, $\hat{f}_i$, and $\hat{f}_l$ are the average spiking rate of excitatory liquid neurons, inhibitory liquid neurons, and all liquid neurons, respectively. Averaged over all NALSM network initializations and both MNIST and N-MNIST datasets, we obtained $f_{E/I} = 0.074 \pm 0.083$, which was indicative of E/I balance since $f_{E/I}$ ranges from 0 to 1, with 0 representing perfect E/I balance. Finally, we

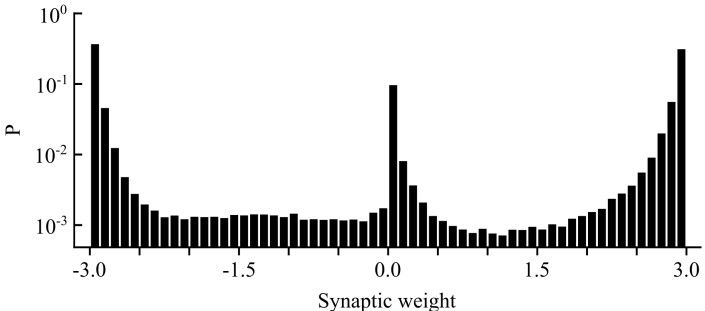

Figure S2: **Initialized NALSM synaptic weights.** Distribution of IL and LL synaptic weights after NALSM liquid initialization.

measured the net current received by each neuron at each timestep from all active input and liquid neurons. Averaged over $100$ different MNIST input samples, net current received by each neuron was $-0.99 \pm 4.91$. The near 0 average net current combined with its large standard deviation suggests that excitatory and inhibitory inputs were balanced and that neurons were primarily driven by network fluctuations. This is believed to give rise to the irregular activity observed in the brain [8] and has been associated with deterministic chaos [6] (shown in Fig. 2 A in [6]).

Indeed, NALSM also exhibited spiking activity that was irregular across the network and across time (Fig. S3). Evidence for chaotic activity was further confirmed by autocorrelation analysis performed on neuronal spike trains generated during generation of spike counts for output layer training (See 2.2.2). Specifically, spike autocorrelation function, $A_{spikes}(\tau_{auto})$, was computed as:

$$A_{spikes}(\tau_{auto}) = \frac{1}{NT} \sum_{i=1}^{N} \sum_{t=1}^{T} \sigma_i(t + \tau_{auto}) \sigma_i(t) \tag{12}$$

where $N = 1000$ liquid neurons, $T = 125 \ ms$ is the duration of neuronal spike trains, $\sigma_i$ is the spike train of neuron $i$. As the branching factor became increasingly greater than $1.0$, the decay of liquid neuron spike autocorrelation functions became broader and increased in magnitude (Fig. S4). Alternatively, when the branching factor became progressively less than $1.0$, decay of liquid neuron spike autocorrelation functions was narrower and magnitudes were marginally greater than that of input neuron spike autocorrelation functions. As expected, spike train autocorrelation functions of input neurons remained flat showing no decay with respect to lag time. This suggested that the transition from a sub-critical to a super-critical branching factor possibly corresponded to a transition to chaos in NALSM's spiking rate dynamics [8, 9].

### A.7  NALSM performance with respect to liquid size

NALSM performance increased with the number of neurons in the liquid, saturating at approximately $8,000$ neurons (Fig. S5)

### A.8  Number of plastic parameters in NALSM

For NALSM, we counted all IL, LL, and LO connections as either plastic with STDP or trainable with gradient descent. For NALSM8000, the number of LO connections was constant at $80,000$. The number of IL, LL connections varied based on the randomly generated liquid. The average number of total plastic/trainable connections for NALSM8000 trained on MNIST was $1,199,406.70 \pm 453.47$ with a maximum(minimum) of $1,200,105(1,198,916)$. For N-MNIST, the average was $3,033,045.40 \pm 268.70$ with a maximum(minimum) of $3,033,446(3,032,737)$. The significant difference in the number of plastic connections used for MNIST and N-MNIST training was due to the $\approx 3$ times larger input layer needed for N-MNIST.

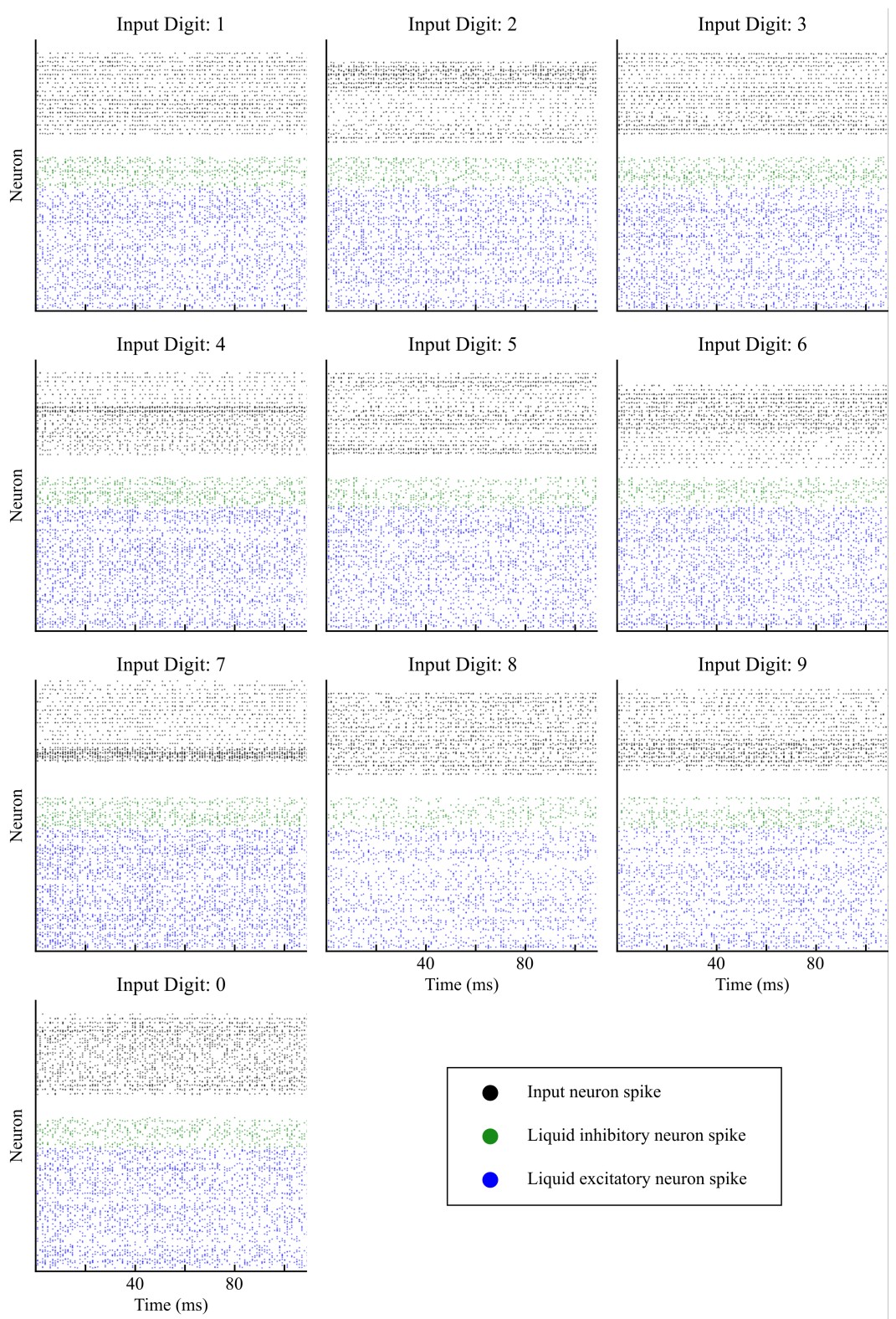

Figure S3: **NALSM network spike activity.** For each input sample class, a raster plot shows spike activity of input (black), liquid inhibitory (green), and liquid excitatory (blue) neurons for a $100\ ms$ duration.

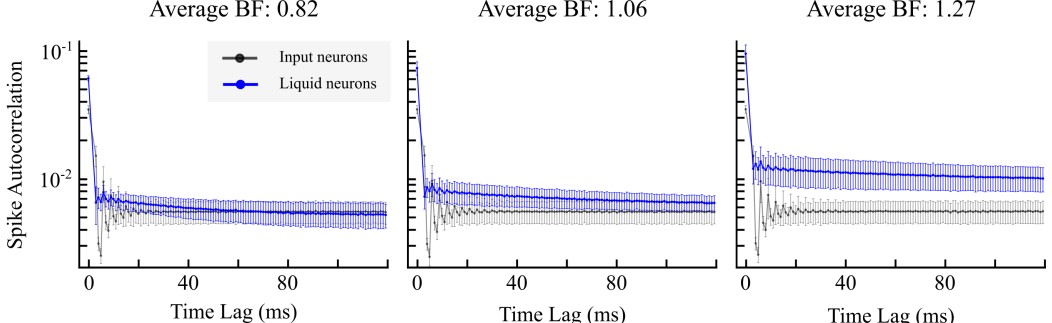

Figure S4: **Spike autocorrelation versus branching factor dynamics.** Spike autocorrelation as a function of lag time for sub-critical (left), near-critical (middle), and super-critical (right) branching factor dynamics. Spike autocorrelation was computed using equation (12) on input (gray) and liquid (blue) neuron spike trains. Data points are averaged over 100 MNIST input samples. Error bars are standard deviation.

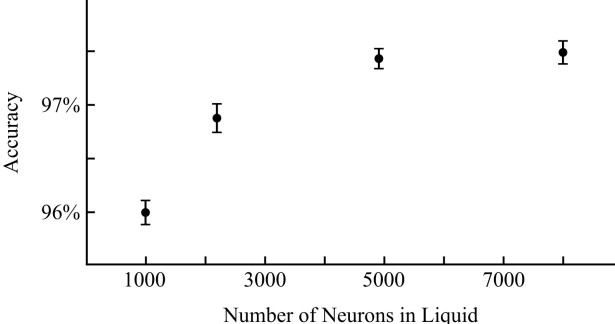

Figure S5: **NALSM accuracy increases with liquid size.** NALSM accuracy shown with respect to number of neurons in the liquid. Data points are averaged over 5 random networks. Error bars are standard deviation.

## A.9 Added computational cost of the LIM astrocyte model

Our proposed method adds a negligible computational cost to the LSM. Specifically, we used a single astrocyte unit with the same functional form as the LIF neuron, making it $0.01\%$ of all the LIF neurons used in NALSM8000 (we used a total of $8,784$ input and liquid neurons for MNIST). In terms of connections, we used $8,784$ neuron-astrocyte connections, which was $0.78\%$ of the number of neuron-neuron connections (we used $1,119,407$ input-liquid and liquid-liquid connections). Further, we showed in Fig. 4 that even with $90\%$ of neuron-astrocyte connections removed, NALSM still maintains a performance advantage versus LSM+AP-STDP and LSM models; in which case only $878$ neuron-astrocyte connections are used or $0.078\%$ of neuron-neuron connections. Finally, fixed neuron-astrocyte connections are computationally less expensive than the plastic neuron-neuron connections, since the ms-precision STDP mechanism (Eqs. 4, 5, 6) adds extra computations on top of each neuronal connection that does not exist in the neuron-astrocyte connections.

## A.10 Curve Fitting

We fit 2nd and 3rd degree polynomial functions. All polynomial fit parameters and residual sum values are shown in Table S2.

## A.11 Hardware

We used Tesla K80 GPU to train all models.

Table S2: Polynomial Curve Fitting Parameters

| Figure/Plot | Degree | Coefficients | Residuals Sum |
|---|---|---|---|
| Fig. 1 B | 3 | $(0.1143, -0.7563, 1.7492, -0.0658)$ | 0.00143 |
| Fig. 2 C MNIST | 2 | $(-0.9151, 2.7587, -0.6077)$ | 0.00229 |
| Fig. 2 C N-MNIST | 2 | $(0.3000, 0.6104, 0.0752)$ | 0.00050 |
| Fig. 4 MNIST | 3 | $(0.0187, -0.0344, 0.0209, 0.9561)$ | 0.000000422 |
| Fig. 4 N-MNIST | 3 | $(0.0044, -0.0084, 0.0060, 0.9585)$ | 0.000000477 |

## Appendices References

[1] Y. Lecun, L. Bottou, Y. Bengio, and P. Haffner. Gradient-based learning applied to document recognition. *Proceedings of the IEEE*, 86(11):2278–2324, 1998. doi: 10.1109/5.726791.

[2] Garrick Orchard, Ajinkya Jayawant, Gregory K. Cohen, and Nitish Thakor. Converting static image datasets to spiking neuromorphic datasets using saccades. *Frontiers in Neuroscience*, 9: 437, 2015. ISSN 1662-453X. doi: 10.3389/fnins.2015.00437.

[3] Han Xiao, Kashif Rasul, and Roland Vollgraf. Fashion-mnist: a novel image dataset for benchmarking machine learning algorithms, 2017.

[4] Yingyezhe Jin and Peng Li. Ap-stdp: A novel self-organizing mechanism for efficient reservoir computing. In *2016 International Joint Conference on Neural Networks (IJCNN)*, pages 1158–1165, 2016. doi: 10.1109/IJCNN.2016.7727328.

[5] Łukasz Kuśmierz, Shun Ogawa, and Taro Toyoizumi. Edge of chaos and avalanches in neural networks with heavy-tailed synaptic weight distribution. *Phys. Rev. Lett.*, 125:028101, Jul 2020. doi: 10.1103/PhysRevLett.125.028101.

[6] C. van Vreeswijk and H. Sompolinsky. Chaos in neuronal networks with balanced excitatory and inhibitory activity. *Science*, 274(5293):1724–1726, 1996. doi: 10.1126/science.274.5293.1724.

[7] Patrick Krauss, Marc Schuster, Verena Dietrich, Achim Schilling, Holger Schulze, and Claus Metzner. Weight statistics controls dynamics in recurrent neural networks. *PLOS ONE*, 14: 1–13, 04 2019. doi: 10.1371/journal.pone.0214541.

[8] Srdjan Ostojic. Two types of asynchronous activity in networks of excitatory and inhibitory spiking neurons. *Nature Neuroscience*, 17:594–600, 2014. ISSN 1546-1726. doi: 10.1038/nn. 3658.

[9] Kanaka Rajan, L. F. Abbott, and Haim Sompolinsky. Stimulus-dependent suppression of chaos in recurrent neural networks. *Phys. Rev. E*, 82:011903, Jul 2010. doi: 10.1103/PhysRevE.82. 011903.