# OpenReview forum: "Increasing Liquid State Machine Performance with Edge-of-Chaos Dynamics Organized by Astrocyte-modulated Plasticity"
_NeurIPS.cc/2021/Conference — NeurIPS 2021 Poster_

### Official Review · Reviewer_879c · 2021-07-15

**Rating:** 7
**Confidence:** 4

**Summary:**

Liquid State Machines (LSM)  circumvents the need for backpropagation and allows the training of recurrent neural networks by local plasticity in the readout weights. Previous studies have shown that optimal performance is achieved near a critical phase transition of the reservoir (the liquid). However, the exact parameters of the critical phase transition in the liquid vary and depend on the input. In this work, the authors present a biologically inspired model that allows self-organization near the critical transition. Here, the authors propose a biologically inspired model in which the liquid self-organizes near the critical point. The model is composed of a reservoir of integrate-and-fire neurons. The synaptic efficacies within have unsupervised spike-time dependent plasticity (STDP). The authors propose an additional dynamical variable akin to the activity of astrocytes in the circuit, which modifies the STDP learning rates. The model self-organizes close to the critical point. The authors train several networks on the MNIST and N-MNIST dataset (a spiking version of MNIST, made for neuromorphic computing). They show that the SLM model with astrocyte-dependent STDP plasticity outperforms other LSM models.

**Limitations And Societal Impact:**

yes

**Main Review:**

### Impact

The main novelty in this work is suggesting that astrocytes may be responsible for the self-organizing of cortical circuits close to criticality. This is an interesting concept and may pave the way for more works studying cortical circuits as liquid state machines.

### Edge of chaos and criticality

It is not clear to me from the paper which critical transition, or edge-of-chaos criticality, the authors imply in the text.

(I) The authors refer to the critical point as the edge-of-chaos in the liquid dynamics (see, e.g., the caption of figure 1). I do not believe that the studied critical point is the onset of chaos of the liquid state, and there are several reasons for that:

1. A spiking model was previously argued not to show a chaotic phase transition (Engelken et al). It was later shown that for spiking, the transition is smoothed to a continuous crossover and the critical effects are strongly diminished (Kadmon and Sompolinsky, PRX, 2015)
2. The connectivity model in eq (3), in which neurons are mostly connected to their neighbors is not known to show chaotic behavior (as far as I know).
3. Strong spatial variance is needed to induce chaos, but it is not clear what the parameter 'C'  is in eq (3). In general, for networks that respect Dale's law, in order to get chaotic behavior, there needs to be a strong balance between excitatory and inhibitory populations [Van Vreeswijk and Sompolinsky 1998]. Alternatively, recent work has studied chaotic dynamics in heavy-tailed distribution [Kuśmierz et al., PRL, 2020]. The authors show neither in their model.
4. There is no evidence that the transition in question is the transition to chaotic activity. The authors use a form of a branching factor (BF) as the order parameter (more on that below). BF>1 leads to critical excitability, and not chaotic behavior.

(II) The authors refer to [22-24] as evidence of edge-of-chaos computation in the brain.  However, these works consider a critical-branching phase transition which is associated with the Mean-Field Directed Percolation (MF-DP) universality class. This is not the transition to chaos (unless in specific settings when the two coincide [see Kuśmierz et al. 2019]. The critical branching-criticality is defined when the branching factor is (BF=1), as implied in this paper. However, it is defined when a single spike (in the liquid!) elicit one spike on average (in the liquid!). The branching factor defined here is different (see my comment below).

(III) Finally the authors may be referring to the edge-of-chaos in feedforward networks, as studied in [Hayou et al. (2019); Lee et al. (2018); Schoenholz et al. (2017)] for rate networks. However, this is not a critical point of the liquid and is not associated with improved computation shown in [19-21]. Furthermore, the edge-of-chaos in the feedforward networks mentioned is defined by the spectrum of the Jacobian of the connectivity matrix, which is not studied in the current paper, so the relation to it is not clear if it exists.

### The use of the branching factor

The order parameter used to measure the dynamical transition in the work is the branching factor (BF). In the proposed model, the dynamics of the astrocytes are driven by a proxy for this measure. There are a few problems I find in this formalism:

1. The criticality in [22-24] is defined with a branching factor of 1 when each spike in the liquid elicits one spike on average within the liquid. This is a different point than the information propagation in feedforward networks [e.g., Hayou et al. (2019); Lee et al. (2018); Schoenholz et al. (2017)]
2. Equations (7) and (8) are not normalized to the size of the input and liquid populations. In the experimental results, the authors change the fraction of liquid neurons that are connected to the astrocyte. Unless I misunderstood, this should change the measure of BFproxy and the astrocyte dynamics. More importantly, is not clear what is the meaning of BF=1 in the case when there is no normalization to the number of neurons sampled.


### Novelty
The paper emphasizes two results: that computation near criticality is better and that astrocyte is a way to get self-organized criticality (SOC). The first was discussed extensively in the past [19-21]. As for SOC, there have been numerous papers suggesting different schemes [e.g., Girardi-Schappo et al, Physical Review Research 2020]. While the idea of using astrocytes as a way to get SOM is interesting, it is not conceptually new in terms of computation, and I feel that this work will fit better in a biologically oriented venue.

**Time Spent Reviewing:**

6

---

> ### Author Response · Authors · 2021-08-10
> **Response to Reviewer 879c**
>
> We thank the Reviewer for their insightful comments. We detail below the responses to their comments and any actions we will take to improve the final version of the manuscript.
>
> $\\\\$
>
> Q3.1.  The critical branching-criticality is defined when the branching factor is (BF=1), as implied in this paper. However, it is defined when a single spike (in the liquid!) elicit one spike on average (in the liquid!). The branching factor defined here is different. The criticality in [22-24] is defined with a branching factor of 1 when each spike in the liquid elicits one spike on average within the liquid.
>
> A3.1. We clarify that we indeed used the branching factor, as defined by the reviewer, and further described on l.177-181, for the purposes of a) measuring liquid dynamics (Fig. 2) and b) validating our proposed astrocyte model’s approximation of the branching factor (Fig.1 B).
>
> $\\\\$
>
> Q3.2.  The connectivity model in eq (3), in which neurons are mostly connected to their neighbors is not known to show chaotic behavior (as far as I know).... Strong spatial variance is needed to induce chaos...
>
> A3.2.  We also clarify that we closely followed the connectivity model used in references [19, 20], where it is shown to exhibit a transition to chaos (measured using Lyapunov exponents).
>
> $\\\\$
>
> Q3.3.  It is not clear to me from the paper which critical transition, or edge-of-chaos criticality, the authors imply in the text... There is no evidence that the transition in question is the transition to chaotic activity. The authors use a form of a branching factor (BF) as the order parameter (more on that below). BF>1 leads to critical excitability, and not chaotic behavior... The authors refer to [22-24] as evidence of edge-of-chaos computation in the brain. However, these works consider a critical-branching phase transition which is associated with the Mean-Field Directed Percolation (MF-DP) universality class. This is not the transition to chaos (unless in specific settings when the two coincide [see Kuśmierz et al. 2019]... In general, for networks that respect Dale's law, in order to get chaotic behavior, there needs to be a strong balance between excitatory and inhibitory populations [Van Vreeswijk and Sompolinsky 1998]. Alternatively, recent work has studied chaotic dynamics in heavy-tailed distribution [Kuśmierz et al., PRL, 2020]. The authors show neither in their model.... Finally the authors may be referring to the edge-of-chaos in feedforward networks, as studied in [Hayou et al. (2019); Lee et al. (2018); Schoenholz et al. (2017)] for rate networks. However, this is not a critical point of the liquid and is not associated with improved computation shown in [19-21]. Furthermore, the edge-of-chaos in the feedforward networks mentioned is defined by the spectrum of the Jacobian of the connectivity matrix, which is not studied in the current paper, so the relation to it is not clear if it exists.
>
> A3.3.  We appreciate the Reviewer's concern and would like to clarify that we used the branching factor (described on l.177-181), as defined in [26] and by the Reviewer, to validate our proposed model’s liquid dynamics (Fig1. B). While references [22-24] in the paper do refer to the critical branching phase transition, this transition has also been found to align with edge-of-chaos (measured using Lyapunov exponent) in biologically plausible recurrent neural networks [1].
>
> To further address the Reviewer’s question, we confirmed that NALSM meets the necessary condition for the two transitions to align, as per Reviewer’s reference [3], by measuring excitation/inhibition (E/I) balance. Specifically, we measured the absolute difference in average excitatory liquid neuron firing rate versus average inhibitory liquid neuron firing rate normalized by the average liquid neuron firing rate. Indicative of E/I balance, we obtained a value of 0.074 (on a scale 0 to 1 with 0 being perfect E/I balance) averaged over all NALSM network initializations and both datasets (average liquid neuron firing rate = 22.5 Hz). We also confirmed E/I balance in initialized NALSM liquid by measuring the difference of positive weights and negative weights normalized by total non-zero weights (weights considered included input-to-liquid and liquid-to-liquid weights) as in [2]. Again indicative of E/I balance, we obtained a value of -0.00882 (on a scale of -1 to 1, with 0 being perfect E/I balance). We will add a section detailing E/I balance calculations and include the above clarifications into our final manuscript.
>
> References:
>
> [1] Haldeman C. and Beggs J. Critical Branching Captures Activity in Living Neural Networks and Maximizes the Number of Metastable States (2005)
>
> [2] Krauss P. et al. Weight statistics controls dynamics in recurrent neural networks (2019)
>
> [3] Van Vreeswijk C. Chaotic Balanced State in a Model of Cortical Circuits (1998)
>
> $\\\\$
>
> Q3.4.  Equations (7) and (8) are not normalized to the size of the input and liquid populations. In the experimental results, the authors change the fraction of liquid neurons that are connected to the astrocyte. Unless I misunderstood, this should change the measure of BFproxy and the astrocyte dynamics. More importantly, is not clear what is the meaning of BF=1 in the case when there is no normalization to the number of neurons sampled.
>
> A3.4.  We thank the Reviewer for highlighting this important point in our neuron-astrocyte connection sparsity ablation study. To clarify, equations (7) (our proposed astrocyte model) and (8) (the heuristic approximated by the astrocyte) do not need normalization, since the astrocyte model effectively measures the balance in aggregate activity between input and liquid neurons. During sampling, we kept the proportion of neurons sampled by the astrocyte the same for both input neurons and liquid neurons, which maintained astrocyte functionality the same as with 100% neuron-astrocyte connectivity. We will add the above mentioned details regarding the neuron-astrocyte connection sparsity ablation study to our final manuscript.
>
> $\\\\$
>
> Q3.5.  It is not clear what the parameter 'C' is in eq (3).
>
> A3.5.  We thank the Reviewer for pointing this out. While we provide the values we used for C in Appendix A.4 and refer to [7] in our paper for more details, we will add a more detailed explanation of parameter C to the final manuscript.
>
> $\\\\$
>
> Q3.6.  The paper emphasizes two results: that computation near criticality is better and that astrocyte is a way to get self-organized criticality (SOC). The first was discussed extensively in the past [19-21]. As for SOC, there have been numerous papers suggesting different schemes [e.g., Girardi-Schappo et al, Physical Review Research 2020]. While the idea of using astrocytes as a way to get SOM is interesting, it is not conceptually new in terms of computation, and I feel that this work will fit better in a biologically oriented venue.
>
> A3.6.  We clarify that while our work does indeed build upon previous findings on “computation near criticality” and “self-organized criticality (SOC)”, its emphasis is on the development of a neuromorphic method that 1) improves the performance of the liquid state machine (LSM) by combining spike-timing-dependent plasticity (STDP) with critical branching(edge-of-chaos) dynamics and 2) minimizes dataset-specific hand-tuning, a pervasive problem of LSMs (discussed on l.268-271).
>
> Existing SOC models, such as the one referenced by the Reviewer [1] and others (See Ref. [26]), focus on developing biologically constrained models that demonstrate how the brain may achieve SOC. Such models have not been demonstrated to work in a machine learning context (as was our target in this work) and often require extensive hand-tuning (which is also addressed by our method).
>
> As our work translates biologically plausible cellular mechanisms to experimentally observed network functions, brain-inspired models like the one we propose here have fit into NeurIPS, especially under the neuroscience and cognitive science category. We hope that the Reviewer will find not only our idea as interesting, but also its implementation as a good fit for this conference.
>
> References:
>
> [1] Girardi-Schappo et al. Synaptic balance due to homeostatically self-organized quasicritical dynamics (2020)

---

> > ### Comment · Reviewer_879c · 2021-08-27
> > **Not convinced about the authors response regarding the transition to chaos; accept their claim for novelty.**
> >
> > **1. criticality and transition to chaos**
> > Despite the authors' explanations, I am not yet convinced that the transition observed is the transition to chaos. First, ref [1] in the rebuttal that they cite for reference does not discuss the chaotic activity. Second, the only model (that I know) in which the transition to chaos and a branching criticality align is random connectivity with power-law distribution [Kuśmierz et al., PRL, 2020]. The heavy-tailed distribution (with unbounded variance) allows the coexistence of strong connections that induce the branching criticality and disorder that induces chaos.
> >
> > In reference [19], where the term "transition to chaos" is used, the network dynamics is binary, and neurons are either "on" or "off". In this dynamics, a positive Lyapunov exponent can lead to either chaos or a frozen state, depending on the connectivity. However, for spiking neurons, as in this work, it is not clear that the dynamics will be chaotic. Moreover, it is likely that that the network will get into fast oscillations mode [Brunel, 2000].
> >
> > Despite my disagreement with the authors regarding chaos, I do agree that the branching criticality has a positive Lyapunov exponent, which I think is the crucial point for this work. My problem is that positive Lyapunov exponent is a necessary but not a sufficient condition for chaos.
> >
> > As per the response on the E/I balance, I do not understand what the authors are trying to prove. E/I balance resulting in chaotic activity [Van Vreeswijk and Sompolinsky, Science 1996] balances the strong mean synaptic inputs needed for $O(1)$ fluctuations when the connectivity is drawn from a normal distribution. I don't see how quantifying the mean E/I balance here supports the claim for chaotic activity. It would have been more beneficial to measure the mean fluctuations of the input across the population [e.g., Ostojic, Nature 2014]
> >
> > To conclude the question of criticality, I don't doubt the presented results and the correctness of the branching, but I still claim it is not the claimed transition to chaos.
> >
> > **2. Normalization**
> > The authors' response to my question is satisfying. I hope the authors improve the explanation in the paper if the paper is accepted for publication.
> >
> > **3. Novelty**
> > I accept the authors' claim regarding novelty. In particular,  I think that  demonstrating that SOC in recurrent networks can improve the computation on a standard dataset will be relevant for some of NeurIPS' audience
> >
> >
> > ## Conclusion
> > I am convinced that the results are novel and interesting enough for NeurIPS. I also think that the results do not depend on whether the dynamics above branching criticality are chaotic or not. However, I can't recommend publishing a manuscript that claims (in the title!) edge-of-chaos dynamics without convincing the correct dynamics. I think the authors should check the dynamic characteristics of their model and how they affect the computation.

---

> > > ### Author Response · Authors · 2021-08-30
> > > **Response 2 to Reviewer 879c**
> > >
> > > We thank the Reviewer for spending their time on our paper and staying open to this discussion. Below we respond to specific comments and detail possible manuscript changes.
> > >
> > > $\\\\$
> > >
> > > Q3.7 As per the response on the E/I balance, I do not understand what the authors are trying to prove. E/I balance resulting in chaotic activity [Van Vreeswijk and Sompolinsky, Science 1996] balances the strong mean synaptic inputs needed for  fluctuations when the connectivity is drawn from a normal distribution. I don't see how quantifying the mean E/I balance here supports the claim for chaotic activity. It would have been more beneficial to measure the mean fluctuations of the input across the population [e.g., Ostojic, Nature 2014]
> > >
> > > A3.7 We appreciate the Reviewers suggestion to further analyze our model’s dynamics. We clarify that we used E/I balance as an additional indication of the existence of chaos, since E/I balance is necessary for chaotic behavior [3]. Next, we will give additional evidence for the existence of chaotic dynamics, based on Reviewer’s reference ([1]: Ostojic, Nature 2014). We measured the firing rate fluctuations of individual liquid neurons and the network spike train autocorrelation in our model. The fluctuations of individual neuron firing rates varied more than in the weak coupling regime (shown in Fig 3 a (left) in [1]), but less than in the strong coupling regime (shown in Fig 3 a (right) in [1]). The liquid spike train autocorrelation as a function of lag time decayed from $0.073 \pm 0.009$ at lag time of $0$ $ms$, to $0.009 \pm 0.002$ at lag time of $6$ $ms$ and then to $0.006 \pm 0.001$ at lag time of $120$ $ms$ (the maximum measured lag time, since our per sample runtime was $250$ $ms$), averaged across $100$ different MNIST input samples. As expected, spike train autocorrelation for input neurons showed no such decay. The decay in the liquid’s spike train autocorrelation combined with fluctuating firing rates in individual liquid neurons suggests that our proposed model’s firing rate dynamics are chaotic [1,2].
> > >
> > > As per the Reviewer’s suggestion, we also measured the net current received by each neuron at each timestep from all active input and liquid neurons. Averaged over $100$ different MNIST input samples, net current received by each neuron was $-0.99 \pm 4.91$. The near $0$ average net current combined with its large standard deviation suggests that excitatory and inhibitory inputs were balanced and that neurons were primarily driven by network fluctuations. This is believed to give rise to the irregular activity observed in the brain [1] (and observed in our model as described in A3.8) and has been associated with deterministic chaos [3] (shown in Fig. 2 A in [3]).
> > >
> > >
> > > References:
> > >
> > > [1] Srdjan Ostojic. Two types of asynchronous activity in networks of excitatory and inhibitory spiking neurons (2014)
> > >
> > > [2] Rajan K. et al. Stimulus-dependent suppression of chaos in recurrent neural networks (2010)
> > >
> > > [3] Van Vreeswijk C. and Sompolinsky H. Chaos in Neuronal Networks with Balanced Excitatory and Inhibitory Activity (1998)
> > >
> > > $\\\\$
> > >
> > > Q3.8 In reference [19], where the term "transition to chaos" is used, the network dynamics is binary, and neurons are either "on" or "off". In this dynamics, a positive Lyapunov exponent can lead to either chaos or a frozen state, depending on the connectivity. However, for spiking neurons, as in this work, it is not clear that the dynamics will be chaotic. Moreover, it is likely that that the network will get into fast oscillations mode [Brunel, 2000].
> > >
> > > A3.8 We thank the Reviewer for their suggestion to look for the fast oscillations mode. We clarify that this is not the case. Specifically, we looked at the spike raster plots and did not find network oscillations: Spiking activity appeared irregular across the network and across time. While we agree with the Reviewer that conclusions based on binary neuron studies may not generalize to spiking neurons, we would like to point out that [19] shows existence of edge-of-chaos in spiking neuron (LIF) networks connected according to distance-based connectivity, which is the same connectivity as in our model. We also present additional evidence for chaotic dynamics in our model in A3.7.
> > >
> > > References:
> > >
> > > [19] Legenstein R. and Maass W. Edge of chaos and prediction of computational performance for neural circuit models. (2007)
> > >
> > > $\\\\$
> > >
> > > Q3.9 Despite the authors' explanations, I am not yet convinced that the transition observed is the transition to chaos. First, ref [1] in the rebuttal that they cite for reference does not discuss the chaotic activity....
> > >
> > > A3.9 We clarify that although the nature of the chaotic activity is not explicitly discussed in [1], it is quantified using Lyapunov exponents which are used as an indicator for the existence of chaos [2].
> > >
> > > References:
> > >
> > > [1] Haldeman C. and Beggs J. Critical Branching Captures Activity in Living Neural Networks and Maximizes the Number of Metastable States (2005)
> > >
> > > [2] Legenstein R. and Maass W. Edge of chaos and prediction of computational performance for neural circuit models. (2007)
> > >
> > > $\\\\$
> > >
> > > Q3.10 ...Second, the only model (that I know) in which the transition to chaos and a branching criticality align is random connectivity with power-law distribution [Kuśmierz et al., PRL, 2020]. The heavy-tailed distribution (with unbounded variance) allows the coexistence of strong connections that induce the branching criticality and disorder that induces chaos.
> > >
> > > A3.10 While our model does not exhibit a heavy-tailed distribution, it has coexistence of strong and weak connections, which are needed to “induce the branching criticality and disorder that induces chaos”. Specifically, our model has a concentration of near-maximum excitatory weights and a much larger concentration of near-zero weights, with weights also covering the full range in between these extremes. Inhibitory weights exhibit the same kind of skewed bimodal distribution.
> > >
> > > $\\\\$
> > >
> > > Q3.11 I am convinced that the results are novel and interesting enough for NeurIPS. I also think that the results do not depend on whether the dynamics above branching criticality are chaotic or not. However, I can't recommend publishing a manuscript that claims (in the title!) edge-of-chaos dynamics without convincing the correct dynamics. I think the authors should check the dynamic characteristics of their model and how they affect the computation.
> > >
> > > A3.11 We hope the Reviewer will appreciate the additional evidence we provide above (A3.7, A3.8, and A3.10) in support of edge-of-chaos dynamics, and we can include the presented analysis with a discussion about these two transitions and their connection to computational performance in the final paper.
> > >
> > > If the Reviewer is still not convinced about the edge-of-chaos dynamics in our model, given that our paper was deemed as "novel and interesting", while its results "do not depend on whether the dynamics above branching criticality are chaotic or not", we are happy to replace “edge-of-chaos” with “critical branching” in the title and throughout the manuscript.
> > >
> > > Given the consensus of all Reviewers on both the validity of our method and the impact of our results, we believe it would be unfortunate not to include this paper in this year's NeurIPS proceedings due to an ambivalent use of a term.

---

> > > > ### Comment · Reviewer_879c · 2021-08-30
> > > > **Aditional response**
> > > >
> > > > I am still not convinced that the model exhibits transition to chaos. First, using autocorrelations to measure chaos in spiking networks is problematic (see [Engelken et al.  2015] as a comment on [Ostojic 2014]. In particular, the autocorrelation function is not a good measure for chaos in the case of spikes. Furthermore, it is not clear that there is a sharp transition to chaos in spiking networks [Engelken et al.  2015, Kadmon and Sompolinsky 2015].
> > > >
> > > > However, I stand by my previous statement that the results are interesting and do not depend on the transition to chaos. Therefore, I am raising my ratings. I hope that if the manuscript is accepted, the authors will include the new evidence that the network is in a cross-over regime between regular and chaotic (though I don't believe there is a phase transition there). I also hope the authors emphasize that this is conjecture, as it is not clear that the network is at the edge of chaos. Unfortunately, there is no way in NeurIPS to review a revised manuscript, but I trust the **area chair**'s judgment.

---

> > > > > ### Author Response · Authors · 2021-08-30
> > > > > **Response 3 to Reviewer 879c**
> > > > >
> > > > > We appreciate the Reviewer's fast response and thank once more the Reviewer for being supportive of our paper. We will make all discussed changes to the manuscript, emphasizing that our model relies on critical branching dynamics, and include the evidence presented in A3.7, A3.8, and A3.10 in support of the conjecture for the possible edge-of-chaos network dynamics.

---

### Official Review · Reviewer_rDJf · 2021-07-16

**Rating:** 7
**Confidence:** 3

**Summary:**

This paper extends liquid state machine (LSM) with a biologically inspired astrocyte model (NALSM) to optimize the neuronal dynamics in LSM at the edge-of-chaos, to improve the accuracy and stability of LSMs. With NALSM, the spiking model can maintain a high accuracy without re-tuning parameters for different datasets. Experiments with MNIST and N-MNIST dataset demonstrate that NALSM outperforms classic LSM and other STDP-based LSM approaches, and most existing spiking neuron models including multi-layer SNNs.

**Ethical Concerns:**

No.

**Limitations And Societal Impact:**

Yes.

**Main Review:**

1. Overall, the idea of borrowing astrocyte mechanisms to build a self-organized dynamic controller for LSM is novel and interesting. The paper is well organized and presented. Although LSM model is currently a narrow topic, the way of thinking is inspiring and can be beneficial to other spiking neuron models.

2. Why the spike-timing-dependent plasticity (STDP) based inter-connection learning is beneficial for LSM models?

3. Discussions and comparisons with other edge-of-chaos modulation approaches can be added to better demonstrate the advantages of NALSM.


**Time Spent Reviewing:**

4

---

> ### Author Response · Authors · 2021-08-10
> **Response to Reviewer rDJf**
>
> We thank the Reviewer for their insightful and constructive feedback. You may find below our detailed responses to the comments.
>
> $\\\\$
>
> Q2.1.  Why the spike-timing-dependent plasticity (STDP) based inter-connection learning is beneficial for LSM models?
>
> A2.1. We appreciate the reviewer’s insightful question. We clarify that STDP was empirically demonstrated to improve liquid state machine performance [17,18]. This effect is attributed to its ability to rewire the liquid based on local correlations, and adheres to a well-known neuroscience concept of Hebbian learning or “neurons wire together if they fire together” [1].
>
> References:
>
> [1] Caporale N. and Dan Y. Spike Timing–Dependent Plasticity: A Hebbian Learning Rule (2008)
>
> $\\\\$
>
> Q2.2.  Discussions and comparisons with other edge-of-chaos modulation approaches can be added to better demonstrate the advantages of NALSM.
>
> A2.2.  We thank the Reviewer for this suggestion and we will improve the overview of edge-of-chaos self-organization mechanisms that we currently give on l.36-48. To emphasize the advantage of our method, we will mention that existing self-organization LSM mechanisms do not incorporate STDP or other forms of plasticity into the LSM (see Ref. [15,16]). Yet, STDP has been shown to improve LSM performance (see Ref. [17,18]). While computational neuroscience works have explored plasticity as a way to organize edge-of-chaos dynamics (see Ref. [26]), these methods have not been extended to neuromorphic machine learning. We hope that the Reviewer will find this line of thought adequate to support the proposed mechanism that combines STDP-modulation with edge-of-chaos.

---

### Official Review · Reviewer_qHbj · 2021-07-16

**Rating:** 6
**Confidence:** 4

**Summary:**

The authors propose of a way modulating the connections in a liquid state machine (LSM) so that it moves towards the edge of chaos (EOC) to maximize its performance for various tasks. They do this using by introducing an additional "astrocyte" that modulates the parameters of STDP in the neuron model -- specifically the depression term. The authors compare their model to other forms of LSMs.


**Limitations And Societal Impact:**

The authors do not discuss limitations of their work.

**Main Review:**

**Update**: The authors response addressed most of my major concerns, and therefore I would like to upgrade my rating.

## Originality:
Modulating the STDP parameters of a network in this particular way to move the network towards the edge of chaos is quite novel to my knowledge. The authors do an adequate discussion of related work. Being able to move the network towards EOC in an automated way using such a relatively simple mechanism is also a very nice feature to have for training liquids.

## Quality:
The approach is well motivated and simple, and the empirical analysis is reasonable.  The authors compare their approach with a wide variety of other published architectures.

Testing the model on further standard benchmarks for e.g. with TIMIT or other speech recognition tasks can make the performance results more convincing.

There are also differences in how STDP is applied between LSM+STDP and NALSM -- in NALSM the STDP continues to adjust weights when collecting spike count samples but not in the other case. Why?

A bit more generally, what's the spiking rate of the liquid? This is a useful metric to see how efficient it could be on neuromorphic hardware and how relatable it is to biology.

A comparison to backpropagation based approaches for setting liquid weights such as [1] could also be made:

[1] Subramoney, Anand, Franz Scherr, and Wolfgang Maass. "Reservoirs learn to learn." arXiv preprint arXiv:1909.07486 (2019).

Minor: how important is it to have distance dependent connectivity?


## Clarity:
While the text does convey the core points well, the writing and structure leave much to be desired. There are various redundant parts and some details are missing in the text. Some examples:
* eqn. 3: C is not defined.
* l.105: What does $ A_{-}^{astro} \rightarrow A_{-} $ mean?
* para starting at l.214 is very redundant with figure caption. Ditto para starting l.285


Other comments:
* l.130: The sentence reads as if all the weights in the liquid are equal.
* I would suggest describing LSM+AP-STDP at least briefly.
* Ditto for how the branching factor is calculated.
* It should be explicitly mentioned that the results in Fig. 2 are without astrocytes.
* Need more details on the kernel quality metric.

Minor:
* In many places, references are made using just the number, which could be ambiguous. For e.g. 158: 2.1 should be Sec. 2.1, l.164 etc.

## Significance:
Being able to automatically move a liquid towards EOC can be quite important to avoid manual tuning often required for liquids. But the authors use a heuristic approach to doing this, which might limit how much it can be built on. But given the empirical results, this method can be extremely useful for practitioners.

**Time Spent Reviewing:**

5

---

> ### Author Response · Authors · 2021-08-10
> **Continuation of Response to Reviewer qHbj**
>
> Q1.9.  Para starting at l.214 is very redundant with figure caption. Ditto para starting l.285
>
> A1.9.  We thank the Reviewer for pointing this out and will remove these redundancies in the final manuscript.
>
> $\\\\$
>
> Q1.10.  l.130: The sentence reads as if all the weights in the liquid are equal.
>
> A1.10.  The Reviewer is correct, in the case of the LSM model, all connections have the same absolute weight as was done in [1].
>
> References:
>
> [1] Zhang W. et al. Information-Theoretic Intrinsic Plasticity for Online Unsupervised Learning in Spiking Neural Networks (2019)
>
> $\\\\$
>
> Q1.11.  I would suggest describing LSM+AP-STDP at least briefly.
>
> A1.11.  We appreciate the Reviewer’s suggestion. While we described the LSM+AP-STDP model in Appendix A.6, we will add it to the main final manuscript.
>
> $\\\\$
>
> Q1.12.  Ditto for how the branching factor is calculated.
>
> A1.12.  We agree and will expand our existing branching factor section (discussed on l.178-181) in the final manuscript.
>
> $\\\\$
>
> Q1.13.  It should be explicitly mentioned that the results in Fig. 2 are without astrocytes.
>
> A1.13.  We appreciate the Reviewer's thorough review of our figures. Indeed, the color code for each model in Fig. 2 is apparently not sufficient to clarify the absence of astrocytes, so we will further add explicit wording to the caption in the final manuscript.
>
> $\\\\$
>
> Q1.14.  Need more details on the kernel quality metric.
>
> A1.14.  We appreciate the Reviewer’s suggestion. We will expand our current description of how we measured kernel quality in Appendix
> A.6 to include more step-by-step implementation details, and move it to the main part of the final manuscript.
>
> $\\\\$
>
> Q1.15.  In many places, references are made using just the number, which could be ambiguous. For e.g. 158: 2.1 should be Sec. 2.1, l.164 etc.
>
> A1.15.  We thank the Reviewer for pointing this out and will address such ambiguities in the final manuscript.

---

> ### Author Response · Authors · 2021-08-10
> **Response to Reviewer qHbj**
>
> We thank the Reviewer for their detailed out comments and insightful suggestions. Below you may find our responses and the description of the associated changes to the manuscript.
>
> $\\\\$
>
> Q1.1.  Being able to automatically move a liquid towards EOC can be quite important to avoid manual tuning often required for liquids. But the authors use a heuristic approach to doing this, which might limit how much it can be built on.
>
> A1.1.  We thank the Reviewer for highlighting the importance of automatic tuning of liquids. We agree that a mathematically rigorous analysis of any proposed model is always preferred compared to the heuristic methods that are currently available in the literature. That is why our method tackles edge-of-chaos in a more global, and therefore disciplined way, which is one of the main motivations of our work. In addition, we hope that the Reviewer will find our proposed model as sufficiently robust to build several extensions increasingly applied to LSMs, including:
>
> 1. Multi-liquid architectures (see Ref. [8,10]) can be constructed using our proposed neuron-astrocyte liquid as a basic building block, because the astrocyte component only relies on the activity of input and liquid neurons.
>
> 2. Different network topologies (see Ref. [74]) can be used, because the proposed astrocyte component is independent of liquid connectivity/topology.
>
> 3. Synaptic plasticity in the liquid can be modified as long as it retains spike-timing-dependent plasticity’s (STDP) separation of potentiation and depression components, which is needed for the astrocyte to directly control the depression learning.
>
> $\\\\$
>
> Q1.2.  Testing the model on further standard benchmarks for e.g. with TIMIT or other speech recognition tasks can make the performance results more convincing.
>
> A1.2.  We agree with the Reviewer and further note that the choice of benchmarking datasets is an open debate in the neuromorphic community. Following the same reasons that most works that we cite here have been applied to these particular datasets, we also chose MNIST and N-MNIST datasets because they i) are the benchmark datasets for most algorithms of this kind, which allows for fair and reproducible comparison among them (Table 1), and ii) cover the range of static and event-based datasets. And while we agree that MNIST in particular may not be challenging enough for complex models such as deep spiking CNNs, complex tasks would introduce the need for changes in the network architecture and make any comparison among the models more ambiguous. We hope that the Reviewer will agree that for our proposed model, MNIST and N-MNIST are challenging enough to demonstrate robust accuracy improvements over similar methods.
>
> However, while our experiments were initially limited to MNIST and N-MNIST datasets, we have now extended the benchmarking of our learning rule to more complex tasks, per the Reviewer’s suggestion. Due to time and computational constraints, we ran additional experiments and evaluated our model on the Fashion-MNIST dataset. The results are consistent with the ones we have already reported and we will include them in the final manuscript. Specifically, NALSM8000 achieved an accuracy of 85.61 $\pm$ 0.18%, averaged over 5 random liquids. This high performance aligns with the results we reported for MNIST and N-MNIST, and it was marginally higher than state-of-the-art spiking methods trained without backpropagation, with a top reported accuracy of 85.31 $\pm$  0.16% [1,2], and comparable to fully-connected multilayer spiking networks trained with backpropagation, with reported accuracies ranging from 84.97 to 87.26% [3,4]. We hope that the Reviewer will find our efforts and the extra results reported here as adequate evidence for this paper. Our next step is to test our model in more complex domains with rich spatiotemporal information, such as speech and EEG data. Testing the model in even more complex tasks is a direction worth pursuing but it is beyond the scope of this paper, as it requires significantly more resources than the ones typically found in an academic lab.
>
> References:
>
> [1] Guo W. et al. Neural Coding in Spiking Neural Networks: A Comparative Study for Robust Neuromorphic Systems (2021)
>
> [2] Hao Y. et al. A Biologically Plausible Supervised Learning Method for Spiking Neural Networks Using the Symmetric STDP Rule (2019)
>
> [3] Zhang W. and Li P. Spike-Train Level Backpropagation for Training Deep Recurrent Spiking Neural Networks (2019)
>
> [4] Ororbia A. and Mali A. Biologically Motivated Algorithms for Propagating Local Target Representations (2018)
>
> $\\\\$
>
> Q1.3.  There are also differences in how STDP is applied between LSM+STDP and NALSM -- in NALSM the STDP continues to adjust weights when collecting spike count samples but not in the other case. Why?
>
> A1.3.  We thank the Reviewer for pointing this out. We clarify that the application of STDP during spike collection in NALSM allows for online adjustment to slight deviations away from the edge-of-chaos dynamics caused by each input sample’s different level of activity, which is known to change network dynamics [1].  We further clarify that keeping STDP active for LSM + STDP tends to saturate weights and produce too much liquid excitability, because of which we kept STDP off during the training phase for the baseline LSM + STDP method. We will revise the methods section to better describe the rationale for the differences pointed out by the Reviewer.
>
> $\\\\$
>
> Q1.4.  A bit more generally, what's the spiking rate of the liquid? This is a useful metric to see how efficient it could be on neuromorphic hardware and how relatable it is to biology.
>
> A1.4.  We clarify that the average firing rate of a liquid neuron ranged from 23.5 Hz to 36.36 Hz for MNIST and 12 Hz to 22.58 Hz for N-MNIST, depending on the input sample. In terms of neuromorphic computing, these firing rates are comparable to the ones reported in existing LSM [1]. It is important to note that these rates also depend on input encoding: Lower input spike rates will ensure lower spike rates in the liquid. Interestingly enough, these frequencies are within reported biological ranges [2]. We will include the above reported rates in the final manuscript.
>
> References:
>
> [1] Zhang W. and Li P. Information-Theoretic Intrinsic Plasticity for Online Unsupervised Learning in Spiking Neural Networks (2019)
>
> [2] Roxin A. et al. On the Distribution of Firing Rates in Networks of Cortical Neurons (2011)
>
> $\\\\$
>
> Q1.5.  A comparison to backpropagation based approaches for setting liquid weights such as [1] could also be made
>
> A1.5.  We appreciate that the Reviewer brought this to our attention. Our proposed method requires only one iteration of generating spikes from the liquid, as opposed to multiple iterations proposed in [1]. While such an approach may be feasible for smaller datasets, it is reasonable to assume that this would become computationally prohibitive for larger datasets. Backpropagation approaches also face severe barriers to deployment on neuromorphic and edge hardware (discussed on l.24-30), which is driving the field to look for brain-inspired solutions. Per the Reviewer’s suggestion, we will briefly discuss a comparison between our method and backpropagation based approaches in the final manuscript.
>
> References:
> [1] Subramoney A. et al. Reservoirs learn to learn (2019)
>
> $\\\\$
>
> Q1.6.  How important is it to have distance dependent connectivity?
>
> A1.6. We thank the Reviewer for the well-thought out question. We used the distance based connectivity originally proposed with the LSM .Generally, connectivity does impact LSM performance (discussed on l.280-282) (see Ref. [74]). Although we demonstrate our proposed astrocyte-modulated STDP mechanism on one type of liquid connectivity, it is reasonable to expect that it can extend to any connectivity, since the mechanism depends only on neuronal spike activity. Therefore, distance based connectivity is not important with respect to our proposed method.
>
> $\\\\$
>
> Q1.7.  eqn. 3: C is not defined.
>
> A1.7.  We thank the Reviewer for pointing this out. While we provide the values we used for C in Appendix A.4 and refer to [7] in our paper for more details, we will add a more detailed explanation of parameter C to the final manuscript.
>
> $\\\\$
>
> Q1.8.  l.105: What does $A_{-}^{astro} \rightarrow A_{-}$ mean?
>
> A1.8.  To clarify, the output of the proposed astrocyte model ($A_{-}^{astro}$) directly maps to the global depression learning rate ($A_{-}$) for liquid-to-liquid and input-to-liquid connections (described on i.105). We will make this clearer in the final manuscript.

---

> > ### Comment · Reviewer_qHbj · 2021-08-27
> > **Note about the benchmarks**
> >
> > For an LSM, I usually find benchmarks that have spatio-temporal structure a lot more useful than those that don't. The latter can easily be solved by feed forward networks, even on neuromorphic hardware.  But I understand that these benchmarks are more useful for comparison with other methods.

---

### Official Review · Reviewer_r3sw · 2021-09-03

**Rating:** 4
**Confidence:** 4

**Summary:**

The manuscript proposes to improve a liquid state machine (implemented in spiking neurons) by adding astrocyte-inspried STDP-mechanism to automatically poise at a 'critical' state, as quantified by a branching factor. The submission claims that close to the so-called 'edge-of-chaos', the proposed networks have improved performance, as quantified in MNIST and N-MNIST.

**Limitations And Societal Impact:**

The presented does not systematically vary various parameters of the proposed NASLM to investigate its impact on both chaoticity and performance so the overall claim that the performance of liquid state machines is improved closed to the edge of chaos using the proposed astrocyte-mediated mechanisms seems not fully supported.
To spell it out in more detail:
* Quantification of chaos (Lyapunov exponent) is missing
* Mathematical theory describing the self-tuning towards an 'edge-of-chaos' is missing
* Mathematical theory linking 'edge of chaos' to superior performance is also missing.

**Main Review:**

Originality:
* liquid state machines (LSM) are a classical alternative to BPTT. Here are novel neuron-astrocyte LSM is being proposed and benchmarked, which seems to be original.

Quality:
* The presented study is a purely numerical work, a mathematical understanding of the neuron-astrocyte LSM is missing.
* The paper claims that the NASLM self-regulate towards the edge of chaos, but this claim is not being supported by standard measures of chaos (e.g. Lyapunov exponents). Instead, a "branching factor" is being computed, which is being interpreted as a proxy of chaos. This claim is not supported by any further evidence.

* Given the numerical nature of the study, important information is missing, e.g. how the baseline was optimized. Moreover, a comparison not only in terms of performance but also in terms of total computational cost (in flops or GPU hours) is missing.

Clarity: The manuscript is written clearly and very understandable.

* significance:
The significance of the presented work seems to be limited, as the NASLM is only benchmarked against other versions of SLM, so it is not clear how it compares to more widely used RNN training methods.


**Time Spent Reviewing:**

5h

---

> ### Author Response · Authors · 2021-09-04
> **Response to Reviewer r3sw**
>
> **Q4.4** The significance of the presented work seems to be limited, as the NASLM is only benchmarked against other versions of SLM, so it is not clear how it compares to more widely used RNN training methods.
>
> **A4.4** Although we agree with the Reviewer that a comparison to other RNN is genuinely an interesting question, we reason below why we included as our baseline only LSM models. Since our goal was to propose a novel self-tunable LSM model, it is reasonable to compare its performance against the state-of-the-art LSMs. Any empirical comparison between an LSM and other RNN methods that are trained via backpropagation of gradients, would entail a significant number of design choices, ranging from the number of neurons to the RNN’s architecture. This would cast doubts on the straight comparison among models, and limit the significance of the proposed astrocytic contribution to the reported results.
>
> To further address the Reviewer’s concern, we provide the computational advantages of using LSMs versus the more widely used RNNs trained via backpropagation. Specifically, we list: a) the computational advantages of LSM (see l.24-30), namely their ability to use depth and recurrence without the downsides associated with backpropagation of gradients, such as vanishing gradients and costly backward passing of information [1,2], and b) the compatibility of LSM with neuromorphic hardware that enables orders of magnitude more energy efficient deployment than that of standard neural networks deployed on GPUs (see l.25-27). Per the Reviewer’s suggestion, we will further discuss the importance of our results in light of the limitations in training the traditional RNNs, emphasizing on a) the easiness of training, and b) computational efficiency (see also A 4.6).
>
> Finally, and for the Reviewer’s convenience, we list below the comments on the significance of the presented work from the other Reviewers, and hope that the Reviewer will find the significance of the presented work as sufficient.
>
> &nbsp;&nbsp;&nbsp;&nbsp;&nbsp;&nbsp; **1) Reviewer qHbj**: “Being able to automatically move a liquid towards EOC can be quite important to avoid manual tuning often required for liquids... this method can be extremely useful for practitioners”. “Modulating the STDP parameters of a network in this particular way to move the network towards the edge of chaos is quite novel to my knowledge..Being able to move the network towards EOC in an automated way using such a relatively simple mechanism is also a very nice feature to have for training liquids.”
>
> &nbsp;&nbsp;&nbsp;&nbsp;&nbsp;&nbsp; **2) Reviewer rDJf**: “The idea of borrowing astrocyte mechanisms to build a self-organized dynamic controller for LSM is novel and interesting. The paper is well organized and presented. Although LSM model is currently a narrow topic, the way of thinking is inspiring and can be beneficial to other spiking neuron models.”
>
> &nbsp;&nbsp;&nbsp;&nbsp;&nbsp;&nbsp; **3) Reviewer 879c**: “This is an interesting concept and may pave the way for more works studying cortical circuits as liquid state machines”. “I accept the authors' claim regarding novelty. In particular, I think that demonstrating that SOC in recurrent networks can improve the computation on a standard dataset will be relevant for some of NeurIPS' audience”. “I am convinced that the results are novel and interesting enough for NeurIPS.”
>
> References:
>
> [1] Ma WD.K. et al. The HSIC Bottleneck: Deep Learning without Back-Propagation (2019)
>
> [2] Lee DH. et al. Difference Target Propagation (2015)
>
> $\\\\$
>
> **Q4.5** Given the numerical nature of the study, important information is missing, e.g. how the baseline was optimized...
>
> **A4.5** We appreciate the Reviewer’s inquiry regarding LSM optimization. As we describe on l.30-32 and l.191-196, we optimized LSM models through the same brute force search of parameters presented in the referenced papers. On l.197-206, we also describe how we trained the LSM by proposing the astrocyte-modulated STDP mechanism to auto-tune liquid weights and minimize brute force search (in addition to increasing accuracy performance). Specifically, we show on Fig. 2 A, B how LSM accuracy performance changes with a liquid’s weight and that our proposed model (NALSM) automatically stabilizes at the necessary dynamics where accuracy peaks without the need to manually tune the liquid weights. Further, we provide all LSM parameters we used in Appendices A.3, A.4 and on l.77-93, l.129-131, and l.146-154.
>
> $\\\\$
>
> **Q4.6** ...Moreover, a comparison not only in terms of performance but also in terms of total computational cost (in flops or GPU hours) is missing.
>
> **A4.6** We clarify that the proposed NALSM model is inherently computationally less expensive compared to both the LSM and LSM+AP-STDP models, due to the fact that NALSM requires one session of training compared to the multiple training sessions for the other LSM methods that need brute force fine tuning. Please also see our discussion in A1.5 and on the manuscript (l.194-196 and l.218-220).
>
> To further address the Reviewer’s concern, we provide below a comparison of the computational costs, which we would be happy to add in the final manuscript. Our method adds a negligible computational cost to the LSM. Specifically, we use $1$ astrocyte unit with the same functional form as the LIF neuron, making it an $0.01\\%$ of all the LIF neurons used in NALSM8000 (we used a total of $8,784$ input and liquid neurons for MNIST). In terms of connections, we used $8,784$ neuron-astrocyte connections, which was an $0.78\\%$ of the number of neuron-neuron connections (we used $1,119,407$ input-liquid and liquid-liquid connections). Further, we showed in Fig.4 that even with $90\\%$ of neuron-astrocyte connections removed, NALSM still maintains a performance advantage versus LSM+AP-STDP and LSM models; in which case only $878$ neuron-astrocyte connections are used or $0.078\\%$ of neuron-neuron connections. Finally, fixed neuron-astrocyte connections are computationally less expensive than the plastic neuron-neuron connections, since the ms-precision STDP mechanism (Eqs. 4, 5, 6)  adds extra computations on top of each neuronal connection that does not exist in the neuron-astrocyte connections. We hope the Reviewer agrees that this demonstrates the computational cost advantages of our method. We can add this analysis to the final manuscript.

---

> ### Author Response · Authors · 2021-09-04
> **Response to Reviewer r3sw**
>
> **Q4.2** The paper claims that the NASLM self-regulate towards the edge of chaos, but this claim is not being supported by standard measures of chaos (e.g. Lyapunov exponents). Instead, a "branching factor" is being computed, which is being interpreted as a proxy of chaos. This claim is not supported by any further evidence….Quantification of chaos (Lyapunov exponent) is missing
>
> **A4.2** We clarify that the branching factor is one of the most widely used metrics for critical network dynamics [1, 2, 3]. We reiterate that the branching factor of the LSM liquid has been found to align with other standard measures of chaos including the Lyapunov exponents in several cases [4, 5]. And, as Reviewer 879c pointed out, “positive Lyapunov exponent is a necessary but not a sufficient condition for chaos”. While we agree that measuring Lyapunov exponents would further validate our proposed method, we explain why this is not possible in our case: Since our proposed mechanism modulates STDP on the millisecond timescale, the measurement of Lyapunov exponents is non-trivial due to the need for fixed synaptic weights/parameters or, at least, enough time-scale separation between activity and plasticity mechanisms to temporarily fixate all parameters. The Reviewer might want to read the discussion and the additional evidence we presented to Reviewer 879c in support of edge-of-chaos dynamics (see A3.3, A3.7, A3.8, A3.10). For the Reviewer’s convenience, we summarize the main points of the discussion below:
>
> &nbsp;&nbsp;&nbsp;&nbsp;&nbsp;&nbsp; **1)** Decay in the spike train autocorrelation with respect to lag time and fluctuations in the firing rates of individual liquid neurons suggests the liquid’s firing rate dynamics are chaotic [6,7]. (A3.7)
>
> &nbsp;&nbsp;&nbsp;&nbsp;&nbsp;&nbsp; **2)** Average net current received by each liquid neuron was near $0$ with high variance, suggesting excitation/inhibition balance and that neuronal activity is driven by fluctuations. This is associated with deterministic chaos [8]. (A3.7)
>
> &nbsp;&nbsp;&nbsp;&nbsp;&nbsp;&nbsp; **3)** Balance of excitatory and inhibitory weights suggested existence of positive Lyapunov exponents [9] (A3.3)
>
> &nbsp;&nbsp;&nbsp;&nbsp;&nbsp;&nbsp; **4)** Our model has coexistence of both weak and strong liquid-to-liquid connection weights, which has been found necessary for critical branching and edge-of-chaos to align [10] (A3.10)
>
> &nbsp;&nbsp;&nbsp;&nbsp;&nbsp;&nbsp; **5)** Spiking activity of liquid neurons appeared irregular across the network and across time. (A3.8)
>
> Finally, being “convinced that the results are novel and interesting enough for NeurIPS”, Reviewer 879c found that “the results do not depend on whether the dynamics above branching criticality are chaotic or not” and agreed “that the branching criticality has a positive Lyapunov exponent, which I think is the crucial point for this work”. Given that Reviewer 879c noted “that positive Lyapunov exponent is a necessary but not a sufficient condition for chaos”, we said that in the final manuscript we will restate edge-of-chaos dynamics as a conjecture, and emphasize that our model exhibits critical branching dynamics. We hope that both our analysis above and our revised approach in better presenting it will help the Reviewer to appreciate the validity and the impact of our work.
>
>
> References:
>
> [1] Beggs J. M. and Plenz D. Neuronal avalanches in neocortical circuits. The Journal of Neuroscience (2003)
>
> [2] Stepp N., et al. Synaptic plasticity enables adaptive self-tuning critical networks. PLOS Computational Biology (2015)
>
> [3] Shriki O., et al. Neuronal Avalanches in the Resting MEG of the Human Brain. The Journal of Neuroscience (2013)
>
> [4] Haldeman C. and Beggs J. Critical Branching Captures Activity in Living Neural Networks and Maximizes the Number of Metastable
> States. Physical Review Letters (2005)
>
> [5] Magnasco M. et al. Self-Tuned Critical Anti-Hebbian Networks. Physical Review Letters (2009)
>
> [6] Srdjan Ostojic. Two types of asynchronous activity in networks of excitatory and inhibitory spiking neurons. Nature Neuroscience (2014)
>
> [7] Rajan K. et al. Stimulus-dependent suppression of chaos in recurrent neural networks. Physical Review E (2010)
>
> [8] Van Vreeswijk C. and Sompolinsky H. Chaos in Neuronal Networks with Balanced Excitatory and Inhibitory Activity. Science (1998)
>
> [9] Krauss P. et al. Weight statistics controls dynamics in recurrent neural networks. PLOS One (2019)
>
> [10] Kusmierz L. et al. Edge of Chaos and Avalanches in Neural Networks with Heavy-Tailed Synaptic Weight Distribution. Physical
> Review Letters (2020)
>
> $\\\\$
>
> **Q4.3** The presented does not systematically vary various parameters of the proposed NASLM to investigate its impact on both chaoticity and performance so the overall claim that the performance of liquid state machines is improved closed to the edge of chaos using the proposed astrocyte-mediated mechanisms seems not fully supported.
>
> **A4.3** We clarify that Fig. 2B, Fig. 2C and Fig. 3A systematically demonstrate improvements in the liquid state machines (LSM) that take place at critical branching dynamics. Specifically, in Fig. 2B, we show how the LSM accuracy changes with liquid dynamics, and demonstrate that NALSM successfully stabilizes near the critical branching factor where LSM accuracy peaks. In Fig. 2C, we show how LSM dynamics change with liquid weight, which we further discuss on l.191-206. In Fig. 3A, we show that it is the combination of critical branching dynamics and STDP that are needed to maximize NALSM performance, and we discuss the implications of our comparative results to prior methods on l.61-62, 271-273, l.277-280. We hope that the Reviewer will find our systematic analysis as sufficient to identify the source of the accuracy improvement for the proposed model.

---

> ### Author Response · Authors · 2021-09-04
> **Response to Reviewer r3sw**
>
> We thank the reviewer for their feedback. We are glad the reviewer found our work to be “original’ and our manuscript “written clearly and very understandable”. Below you may find our detailed responses to the comments.
>
> **Q4.1** The presented study is a purely numerical work, a mathematical understanding of the neuron-astrocyte LSM is missing….Mathematical theory describing the self-tuning towards an 'edge-of-chaos' is missing….Mathematical theory linking 'edge of chaos' to superior performance is also missing.
>
> **A4.1** We agree with the Reviewer that a rigorous mathematical analysis is preferred to an empirical study, in cases where a theoretical study is feasible. That said, an increasingly large number of papers published at NeurIPS are empirical [1, 2, 3, 4, 5, 6]. In fact, the deep learning literature is based on seminal empirical papers [7, 8, 9, 10, 11].
>
> In addition, the link between 'edge of chaos' and superior performance has long been established: The computational performance of systems poised at the critical phase transition has been widely studied both experimentally [12] and theoretically [13], and are well-connected to critical branching transition [14,15] as well as edge-of-chaos transition (for a rigorous mathematical analysis see [16,17]). Theoretical studies have also shown that by maintaining near-critical dynamics, networks can simultaneously access the computational properties (learning and memory) of both phases and, thereby, 1) maximize their information processing capacity [12], 2) optimize their dynamical range [18,19], and 3) expand their number of metastable states [14]. Studies on reservoir performance with respect to criticality have further suggested significant learning benefits including improved accuracy and generalization capabilities [16,17]. Nevertheless, while our proposed method is heuristic in nature, we describe on l.108-121 how it corresponds to the branching factor and show in Fig. 1 B that our proposed heuristic and the branching factor metric align. For the extensibility of our heuristic approach, please also see our discussion with Reviewer qHbj (A1.1).
>
> To address the Reviewer’s concern, and if the Reviewer finds it useful, we would be happy to include a summary of our response in the final manuscript.
>
> References:
>
> [1] Bellec G., et al. Long short-term memory and learning-to-learn in networks of spiking neurons. NIPS (2018)
>
> [2] Antoniou A. and Storkey A. Learning to Learn via Self-Critique. NIPS (2019)
>
> [3] Yeh R. A., et al. Chirality Nets for Human Pose Regression. NIPS (2019)
>
> [4] Ma B., et al. Auto Learning Attention. NIPS (2020)
>
> [5] Chen L., et al. Trading Personalization for Accuracy: Data Debugging in Collaborative Filtering. NIPS (2020)
>
> [6] Guo Y., et al. Backpropagating Linearly Improves Transferability of Adversarial Examples. NIPS (2020)
>
> [7] Le Cun Y., et al. Handwritten Digit Recognition with a Back-Propagation Network. NIPS (1989)
>
> [8] Krizhevksy A., et al. ImageNet Classification with Deep Convolutional Neural Networks. NIPS (2012)
>
> [9] Mikolov T., et al. Distributed Representations of Words and Phrases and their Compositionality. NIPS (2013)
>
> [10] Mnih V. et al. Playing Atari with Deep Reinforcement Learning. NIPS (2013)
>
> [11] Sabour S., et al. Dynamic Routing Between Capsules. NIPS (2017)
>
> [12] Shew W.L., et al. Information capacity and transmission are maximized in balanced cortical networks with neuronal avalanches. The Journal of Neuroscience (2011)
>
> [13] De Arcangelis L. and Herrmann H.J. Learning as a phenomenon occurring in a critical state. PNAS (2010)
>
> [14] Haldeman C. and Beggs J. Critical Branching Captures Activity in Living Neural Networks and Maximizes the Number of Metastable States. Physical Review Letters (2005)
>
> [15] Balafrej I. and Rouat J. P-critical: A reservoir autoregulation plasticity rule for neuromorphic hardware. Arxiv (2020)
>
> [16] Legenstein R. and Maass W. Edge of chaos and prediction of computational performance for neural circuit models. Neural Networks (2007)
>
> [17] Bertschinger N. and Natschlager T. Real-time computation at the edge of chaos in recurrent neural networks. Neural Computation (2004)
>
> [18] Kinouchi, O. and Copelli, M. Optimal dynamical range of excitable networks at criticality. BMC Neuroscience (2006).
>
> [19] Shew, W. L., et al. Neuronal avalanches imply maximum dynamic range in cortical networks at criticality. The Journal of Neuroscience (2009).

---

### Author Response · Authors · 2021-08-10
**We thank all the reviewers for their time and thoughtful feedback**

We thank the Reviewers for their time and insightful feedback in reviewing our paper. We are encouraged that they found our work "important", "well motivated and simple" (Reviewer qHbj), "novel and interesting" and that our "way of thinking is inspiring and can be beneficial to other spiking neuron models" (Reviewer rDJf). We are pleased that the manuscript was deemed "well organized and presented" (Reviewer rDJf), and that our method was favorably reviewed as "extremely useful for practitioners" (Reviewer qHbj), which "may pave the way for more works studying cortical circuits as liquid state machines" (Reviewer 879c). We answer specific questions below and detail the changes in the manuscript upon incorporating the feedback.

---

### Decision · Program_Chairs · 2021-09-28

**Decision:**

Accept (Poster)

**Comment:**

The reviewers agree that the main contribution of this paper, i.e. that the hypothesis that astrocytes help networks to self-organize close to criticality, is novel. However, the concrete impact of this contribution is somewhat unclear. Is this a statement about the function of astrocytes in biological neural circuits? If so, the paper would ideally argue in greater detail that their astrocytes model is realistic, and furthermore that other spike-rate adaptation mechanisms do not have similar, beneficial effects. Conversely, as a contribution to algorithms on neuromorphic hardware for tackling practical problems, this work would benefit from additional comparisons to alternatives for introducing stabilizing negative feedback mechanisms. Therefore, the overall contribution of this paper is below acceptance.



**Consistency Experiment:**

NeurIPS has a long history of experimentation. In 2014, NeurIPS ran an experiment in which 10% of submissions were reviewed by two independent committees to quantify the randomness in the review process. This year, we repeated a variant of this experiment to see how the quality of the review process has changed over time.  This paper was part of the experiment and was therefore assigned to two committees (consisting of reviewers, an Area Chair, and a Senior Area Chair) that reached independent decisions.  If both committees made the same recommendation, this recommendation was followed. If a single committee recommended acceptance, the paper was accepted (with the exception of a few cases in which the other committee identified what we considered a fatal flaw, e.g., an error in a key result).

This copy’s committee reached the following decision: **Reject**

The other committee assigned to the paper recommended **Accept (Poster)**.  You can find the other set of reviews, along with any follow up discussion with the authors here:
https://openreview.net/forum?id=0isj8oxdQys